# Sexual dimorphic regulation of recombination by the synaptonemal complex in *C. elegans*

**Cori K Cahoon, Colette M Richter, Amelia E Dayton, Diana E Libuda***

Institute of Molecular Biology, Department of Biology, University of Oregon, Eugene, United States

**Abstract** In sexually reproducing organisms, germ cells faithfully transmit the genome to the next generation by forming haploid gametes, such as eggs and sperm. Although most meiotic proteins are conserved between eggs and sperm, many aspects of meiosis are sexually dimorphic, including the regulation of recombination. The synaptonemal complex (SC), a large ladder-like structure that forms between homologous chromosomes, is essential for regulating meiotic chromosome organization and promoting recombination. To assess whether sex-specific differences in the SC underpin sexually dimorphic aspects of meiosis, we examined *Caenorhabditis elegans* SC central region proteins (known as SYP proteins) in oogenesis and spermatogenesis and uncovered sex-specific roles for the SYPs in regulating meiotic recombination. We find that SC composition, specifically SYP-2, SYP-3, SYP-5, and SYP-6, is regulated by sex-specific mechanisms throughout meiotic prophase I. During pachytene, both oocytes and spermatocytes differentially regulate the stability of SYP-2 and SYP-3 within an assembled SC. Further, we uncover that the relative amount of SYP-2 and SYP-3 within the SC is independently regulated in both a sex-specific and a recombination-dependent manner. Specifically, we find that SYP-2 regulates the early steps of recombination in both sexes, while SYP-3 controls the timing and positioning of crossover recombination events across the genomic landscape in only oocytes. Finally, we find that SYP-2 and SYP-3 dosage can influence the composition of the other SYPs in the SC via sex-specific mechanisms during pachytene. Taken together, we demonstrate dosage-dependent regulation of individual SC components with sex-specific functions in recombination. These sexual dimorphic features of the SC provide insights into how spermatogenesis and oogenesis adapted similar chromosome structures to differentially regulate and execute recombination.

*For correspondence:
dlibuda@uoregon.edu

**Competing interest:** The authors declare that no competing interests exist.

## Editor's evaluation

This important article describes sex- and recombination-dependent dynamics of proteins in a meiosis-specific chromosome structure, the synaptonemal complex. The authors provide solid evidence for their conclusion by cytological analysis with proper quantification. The study is of great interest to researchers in the field of meiosis and chromosomes.

## Introduction

In most sexually reproducing organisms, meiosis ensures the faithful inheritance of genetic information in each generation through the formation of haploid gametes, such as eggs and sperm. Many aspects of meiosis are sexually dimorphic from the differences in the size of egg and sperm cells to the molecular mechanisms ensuring accurate segregation of the chromosomes (reviewed in *Cahoon and*

*Libuda, 2019*). However, many meiotic proteins are present in both sexes and how each sex utilizes a highly similar proteome to produce dimorphic phenotypes remains unclear.

Multiple studies in mice and plants indicate that meiotic chromosome structures, such as the synaptonemal complex (SC), are sexually dimorphic (reviewed in *Cahoon and Libuda, 2019*; *Morgan et al., 2017*). The SC assembles between homologous chromosomes in early prophase I (called the transition zone in *Caenorhabditis elegans* or late zygotene in mice, plants, and yeast) and organizes the genome to both facilitate and enable the essential meiotic processes of homolog pairing and recombination (reviewed in *Cahoon and Hawley, 2016*). This scaffold between the homologs formed by the SC allows for the accurate repair of DNA double-strand breaks (DSBs) induced by the topoisomerase-like protein Spo11 (*Keeney et al., 1997*; *Zickler and Kleckner, 1999*; *Zickler and Kleckner, 2015*). A subset of these DSBs must be repaired as crossover recombination events to allow for the accurate segregation of the homologs at anaphase I.

The SC is both required for the formation of crossovers and influences the frequency of crossovers occurring per homolog pair in multiple organisms (*Colaiácovo et al., 2003*; *Costa et al., 2005*; *de Vries et al., 2005*; *Gordon et al., 2021*; *Hayashi et al., 2010*; *Higgins et al., 2005*; *Hillers and Villeneuve, 2003*; *Hurlock et al., 2020*; *Jeffress et al., 2007*; *Libuda et al., 2013*; *Liu et al., 2021*; *MacQueen et al., 2002*; *Page and Hawley, 2001*; *Smolikov et al., 2007a*; *Smolikov et al., 2009*; *Sym et al., 1993*; *Woglar and Villeneuve, 2018*). Proteins within the SC can influence crossover distribution, DSB repair mechanisms, and the crossover licensing process (*Capilla-Pérez et al., 2021*; *Durand et al., 2022*; *Garcia-Muse et al., 2019*; *Gordon et al., 2021*; *Jeffress et al., 2007*; *Láscarez-Lagunas et al., 2022*; *Voelkel-Meiman et al., 2019*; *Voelkel-Meiman et al., 2015*; *Voelkel-Meiman et al., 2022*). In *C. elegans*, pro-crossover proteins are recruited to the SC by the central region proteins of the SC (*Cahoon et al., 2019*; *Libuda et al., 2013*). Notably, the regulation of recombination is sexually dimorphic in many organisms (*Arbeithuber et al., 2015*; *Bhérer et al., 2017*; *Brick et al., 2018*; *Choi et al., 2018*; *Clément and Arndt, 2013*; *de Boer et al., 2015*; *Drouaud et al., 2007*; *Durand et al., 2022*; *Giraut et al., 2011*; *Gruhn et al., 2013*; *Halldorsson et al., 2019*; *Halliwell and Hoffmann, 2021*; *Morelli and Cohen, 2005*; *Pratto et al., 2021*; *Underwood et al., 2018*). The exact mechanism(s) of how the SC regulates crossing over and obtains sex-specific outcomes remains an active area of study in multiple organisms.

Similar to other organisms, many aspects of *C. elegans* meiosis are sexually dimorphic from the timing of egg and sperm development to the regulation of checkpoints and recombination (*Cahoon and Libuda, 2021*; *Checchi et al., 2014*; *Gartner and Engebrecht, 2022*; *Gumienny et al., 1999*; *Jaramillo-Lambert et al., 2007*; *Jaramillo-Lambert and Engebrecht, 2010*; *Jaramillo-Lambert et al., 2016*; *Jaramillo-Lambert et al., 2010*; *Lamelza and Bhalla, 2012*; *Li et al., 2020*; *Rourke and Jaramillo-Lambert, 2022*; *Saito and Colaiácovo, 2017*). These sex-specific differences suggest that other critical recombination regulatory processes, such as the SC, may also have sexually dimorphic features in *C. elegans*. Both sexes in *C. elegans* are assumed to assemble the same proteins into the SC, but this aspect has not been extensively investigated as most of our knowledge about the SC in *C. elegans* focuses on oocytes.

Here we show that the SC is sexually dimorphic in *C. elegans*. Specifically, we demonstrate that the composition of the SC is not uniform during prophase I and instead is regulated in a sex-specific and protein dosage-dependent manner to facilitate specific steps of recombination. We find that a threshold level of SYP-2 in the SC is critical for the establishment and/or stabilization of recombination intermediates, while SYP-3 levels in the SC modulate the timing of crossover designation during pachytene. In addition, we identify sexual dimorphic regulation of SC composition whereby specific SC proteins independently influence the levels of other proteins within the complex. Taken together, our study reveals novel regulation of recombination whereby the SC composition is dynamically altered throughout pachytene to facilitate sexually dimorphic mechanisms of DNA repair.

## Results

### SYP-2 and SYP-3 are sexually dimorphic

To understand the relationship between the sexually dimorphic aspects of meiosis and the SC, we used the model system *C. elegans* where oocyte and spermatocyte development can be easily accessed and analyzed at both the same time and developmental stage. Adult male worms undergo

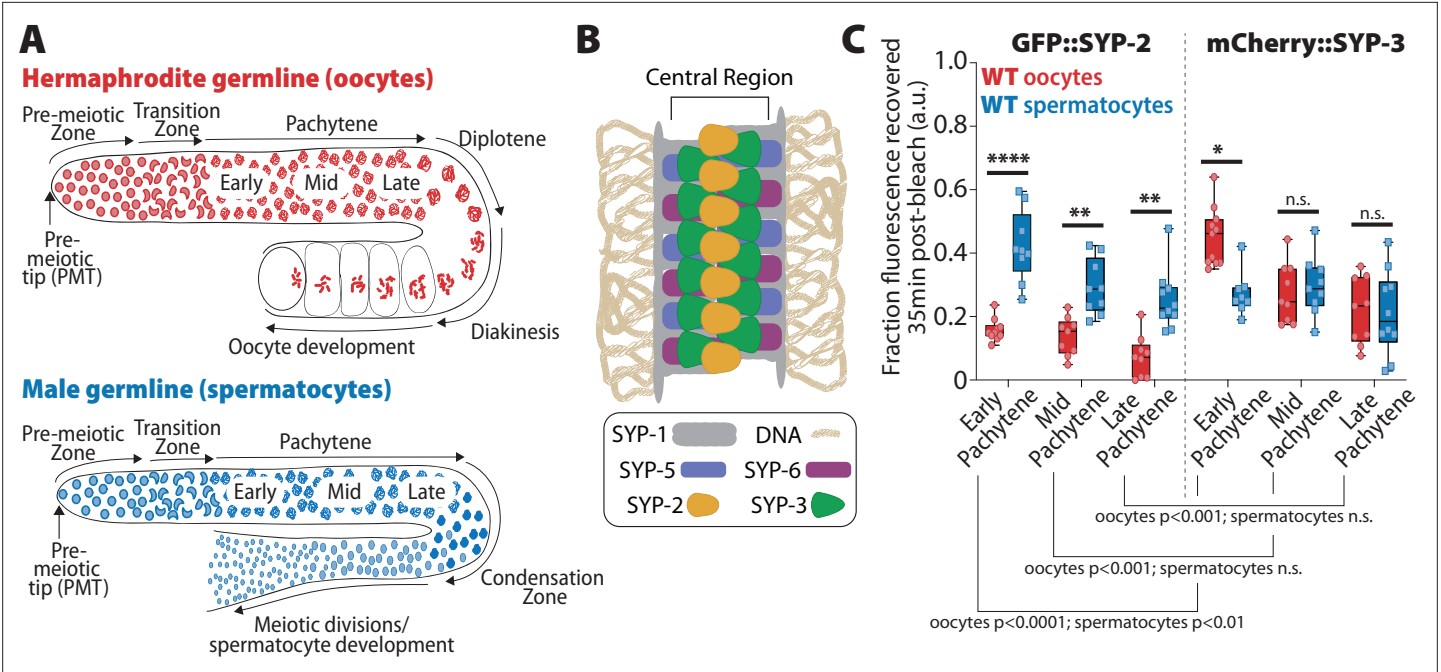

**Figure 1.** SYP-2 and SYP-3 dynamics are sexually dimorphic. (**A**) Diagrams of hermaphrodite (top, red) and male (bottom, blue) germlines with developing oocytes and spermatocytes, respectively. The stages of the germlines are labeled starting at the pre-meiotic tip (PMT) and ending at the meiotic divisions. Nuclei proliferate at the distal end of the germline (pre-meiotic tip) and physically move proximally as they proceed into the stages of meiosis: the transition zone (leptotene/zygotene), pachytene, diplotene, and diakinesis (in spermatocytes diplotene/diakinesis is termed the condensation zone). At the end of prophase I, oocyte nuclei arrest at diakinesis until they are fertilized, but spermatocytes rapidly complete the meiosis I and meiosis II divisions to generate mature sperm. (**B**) Diagram of the synaptonemal complex showing the positions of SYP-1 (gray), SYP-2 (yellow), SYP-3 (green), SYP-5 (blue-purple), and SYP-6 (red-purple) within the central region of the complex. (**C**) Quantification of the fraction of fluorescence recovered 35 min after photobleaching a small region of either GFP::SYP-2 (left) or mCherry::SYP-3 (right) in both oocytes (red)and spermatocytes (blue) (a.u. = arbitrary units). All statistics are multiple comparisons Bonferroni–Dunn adjusted Mann–Whitney $U$ test and unless the p-value is indicated the asterisk indicates number of significant digits from p=0.05 (n.s. = not significant). Oocyte data with GFP::SYP-2 is from 10 nuclei (early), 9 nuclei (mid), and 9 nuclei (late), and with mCherry::SYP-3 is from 11 nuclei (early), 9 nuclei (mid), and 9 nuclei (late). Spermatocyte data with GFP::SYP-2 is from 9 nuclei (early), 9 nuclei (mid), and 10 nuclei (late), and with mCherry::SYP-3 is from 8 nuclei (early), 9 nuclei (mid), and 10 nuclei (late).

The online version of this article includes the following source data and figure supplement(s) for figure 1:

**Source data 1.** The normalized fluorescence recovery for the fluorescent recovery after photobleaching (FRAP) data for each nucleus analyzed with GFP::SYP-2 and mCherry::SYP-3 at early, mid, and late pachytene in both sexes (*Figure 1C*).

**Figure supplement 1.** Montages of fluorescent recovery after photobleaching (FRAP) from oocytes and spermatocytes.

**Figure supplement 2.** Fluorescent recovery after photobleaching (FRAP) recovery curves from oocytes and spermatocytes.

**Figure supplement 3.** Synaptonemal complex (SC) lengths are not different between the sexes in early and mid pachytene.

**Figure supplement 3—source data 1.** The SYP-1 length measurements for each chromosome in wild-type oocytes and spermatocytes.

spermatogenesis, while adult hermaphrodite worms undergo oogenesis (*Figure 1A*). The germline for both sexes is organized as a spatial-temporal gradient along the distal-proximal axis, thereby allowing for easy and simultaneous access to all stages of meiotic prophase I (*Gartner and Engebrecht, 2022*; *Hillers et al., 2017*). The SC initiates assembly late in the transition zone (leptotene/zygotene) and is fully assembled by pachytene (*MacQueen et al., 2002*). While many proteins have been identified within the *C. elegans* SC, we focused our analyses here on the central region proteins, which are all called SYP (SYnaPsis protein) and, to date, six SYP proteins (SYP-1–SYP-6) span the gap between the homologous chromosomes (*Figure 1B*; *Colaiácovo et al., 2003*; *Hurlock et al., 2020*; *MacQueen et al., 2002*; *Smolikov et al., 2007a*; *Smolikov et al., 2009*; *Zhang et al., 2020*).

To assess if SC dynamics differ between spermatocytes and oocytes, we performed fluorescent recovery after photobleaching (FRAP) assays with two SYP proteins endogenously tagged with fluorescent proteins: (1) GFP::SYP-2 from *Gao et al., 2016*, and (2) mCherry::SYP-3 that we generated using CRISPR/Cas9 (see 'Methods'; *Figure 1C*, *Figure 1—figure supplements 1 and 2*). Both SYP-2

(213 amino acids) and SYP-3 (224 amino acids) are the smallest SYP proteins in the *C. elegans* SC and have very little sequence similarity between themselves and the other SYP proteins (*Kursel et al., 2021*). However, structurally SYP-2 and SYP-3 are similar to the other SYP proteins with all the SYP proteins having predicted coiled-coil protein domains (*Kursel et al., 2021*). We found that both SYP-2 and SYP-3 have sex-specific differences in protein turnover, with SYP-2 dynamics higher in spermatocytes and SYP-3 dynamics higher in oocytes.

For GFP::SYP-2, spermatocytes displayed significantly higher recovery dynamics throughout pachytene compared to oocytes (*Figure 1C*, *Figure 1—figure supplements 1 and 2*; p<0.001, Bonferroni–Dunn adjusted, Mann–Whitney). In contrast, mCherry::SYP-3 recovered more quickly in oocytes during early pachytene compared to spermatocytes (early pachytene p=0.0015; Bonferroni–Dunn adjusted, Mann–Whitney). During mid and late pachytene, SYP-3 recovery rates were similar between the sexes. Notably, we observed that the progressive stabilization of spermatocyte SYP-3 was less pronounced and more subtle than that of oocyte SYP-3, suggesting that spermatocytes may not modulate SYP-3 stability via the same mechanisms as oocytes (*Figure 1C*). Collectively, these data indicate that SYP dynamics are not uniformly regulated throughout prophase and instead exhibit both SYP-specific and sex-specific dynamics.

Overall, both sexes showed the same overall trend of progressive stabilization of SYP-2 and SYP-3 throughout pachytene, matching previous observations with the transgene GFP::SYP-3 (*Figure 1*, *Figure 1—figure supplements 1 and 2*; see 'Methods'; *Nadarajan et al., 2017*; *Pattabiraman et al., 2017*; *Rog et al., 2017*). Upon comparing SYP mobilization between the sexes, we found that SYP-2 and SYP-3 displayed differences in mobility when compared to each other within each sex. In oocytes, GFP::SYP-2 turnover was significantly reduced compared to mCherry::SYP-3 turnover (*Figure 1C*, *Figure 1—figure supplements 1 and 2*; p<0.01, Bonferroni–Dunn adjusted, Mann–Whitney). Comparatively in spermatocytes, only early pachytene displayed significant differences in GFP::SYP-2 and mCherry::SYP-3 turnover (*Figure 1C*; p=0.0237, Bonferroni–Dunn adjusted, Mann–Whitney). These results demonstrate that SYP-2 and SYP-3 can be independently regulated during pachytene.

To determine whether differential regulation of SYP-2 and SYP-3 might be influenced by chromosome length, we measured SC length during early, mid, and late pachytene in both sexes. Our results found that SC length was similar throughout pachytene in each sex (*Figure 1—figure supplement 3*). The slightly shorter SC lengths in late pachytene spermatocytes were likely due to differences in chromosome compaction between the sexes (*Rourke and Jaramillo-Lambert, 2022*; *Samson et al., 2014*). These results suggest that independent regulation of SYP-2 and SYP-3 within an assembled SC is not due to changes in chromosome length, but rather by other factors.

## SYP-2 and SYP-3 composition within the SC is sexually dimorphic

The differences in SYP-2 and SYP-3 mobility between oogenesis and spermatogenesis suggest that the abundance or concentration of each protein within the SC may also be sexually dimorphic. To compare the relative assembled SYP compositions within the SC between sexes during meiotic prophase I, we calculated the mean fluorescence intensity per cubic micrometer of assembled SC for individual nuclei throughout pachytene in oogenesis and spermatogenesis, and then calculated the average SYP intensity among nuclei in a sliding window across the normalized germline length (see 'Methods'). We found that the accumulation of GFP::SYP-2 in the SC is sexually dimorphic (*Figure 2A and B*). Wild-type oocytes progressively accumulated SYP-2 up until the early to mid pachytene transition (early pachytene mean intensity 153,292.27 ± SD 13,983.85; early/mid pachytene mean peak intensity 161,878.14± SD 13,930.43; mid pachytene mean intensity 148,430.1± SD 13,058.02) and then reduce the amount of SYP-2 slightly before maintaining a relatively constant level of SYP-2 through late pachytene. In contrast to oocytes, wild-type spermatocytes progressively loaded SYP-2 into the SC until the onset of late pachytene at which point the amount of SYP-2 began to stabilize (*Figure 2A and B*). Notably, early pachytene spermatocytes had less SYP-2 loaded than oocytes, but late pachytene spermatocytes had near or slightly more amounts of SYP-2 in the SC than oocytes (p<0.001, Bonferroni adjusted, Mann–Whitney). Additionally, slightly past the early/mid pachytene transition of SYP-2 intensity, we noted only oocytes display a few persisting bright nuclei in nearly all germlines examined (eight out of nine germlines with persisting bright nuclei; *Figure 2B*). Thus, the incorporation of SYP-2 throughout pachytene differs between sexes.

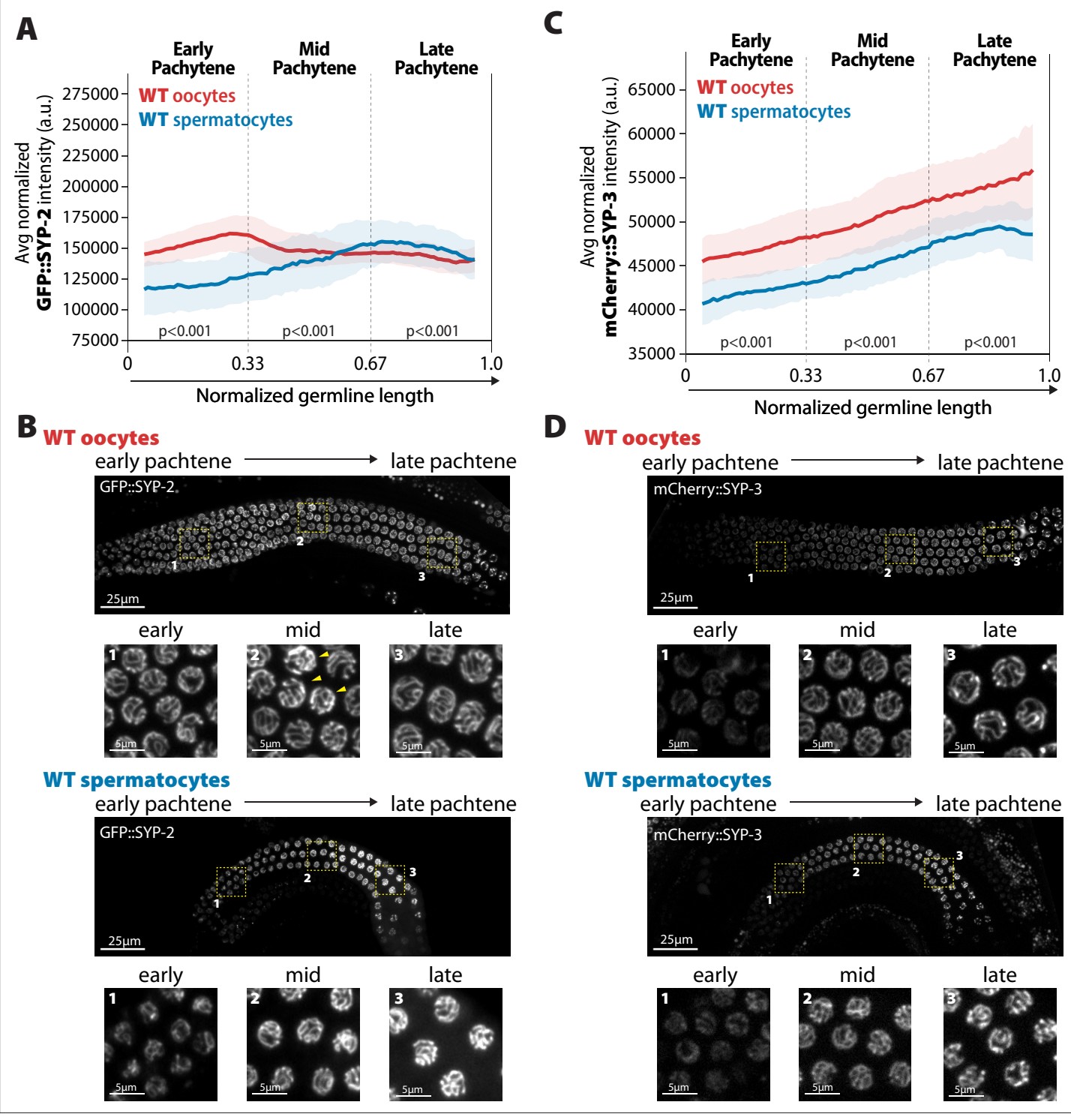

**Figure 2.** Accumulation of SYP-2 and SYP-3 in the synaptonemal complex (SC) is sexually dimorphic. (**A,C**) Quantification of the mean intensity of GFP::SYP-2 (**A**) or mCherry::SYP-3 (**C**) per nucleus normalized by the volume of each nucleus (see 'Methods') throughout pachytene for wild-type (WT) oocytes (red, pale red band is the standard deviation) and spermatocytes (blue, pale blue band is the standard deviation). p-Values on the plot are comparisons between oocytes and spermatocytes for each region of pachytene using Mann–Whitney *U* tests. (**B, D**) Represented images of GFP::SYP-2 (**B**) or mCherry::SYP-3 (**D**) in WT hermaphrodite (top, oocytes) and male (bottom, spermatocytes) germlines. Germlines are oriented with the start of pachytene on the left and meiotic progression continues to the right. Yellow boxes identify the regions enlarged in each image below to show representatives of early, mid, and late pachytene. Arrowheads indicate GFP::SYP-2 bright nuclei in mid pachytene. The intensity adjustments are

*Figure 2 continued on next page*

*Figure 2 continued*

the same for both GFP::SYP-2 germlines and mCherry::SYP-3 germlines, respectively. n values for the number of germlines and nuclei can be found in *Figure 2—source data 2*.

The online version of this article includes the following source data for figure 2:

**Source data 1.** Raw sum intensity and normalized sum intensity per nucleus for GFP::SYP-2, mCherry::SYP-3 in both sexes and wild-type genotypes (*Figure 2A,C*).

**Source data 2.** SC intensity n values for nuclei and germlines scored.

In contrast to SYP-2, mCherry::SYP-3 progressively accumulated throughout pachytene in both sexes, matching previous observations using a GFP::SYP-3 transgene (*Figure 2C and D*; *Pattabiraman et al., 2017*). Therefore, differences in fluorescent tags on SYP-3 do not appear to largely influence the incorporation of SYP-3 into the SC during pachytene (see 'Methods'). In comparison to oocytes, spermatocytes incorporated less SYP-3 throughout pachytene (p<0.001, Bonferroni adjusted, Mann–Whitney), which was also observed in the early regions of pachytene with SYP-2. Thus, spermatocytes have less SYP-3 and SYP-2 in the SC than oocytes, specifically within the early and mid regions of pachytene. Moreover, unlike with SYP-2, we did not observe in either sex any bright SYP-3 nuclei that were not surrounded by other nuclei of similar intensity, thereby suggesting that these bright SYP-2 nuclei may have defects or changes that only trigger a response with SYP-2 levels (*Figure 2D*). Taken together, these data demonstrate that SYP-2 and SYP-3 are differentially incorporated in the SC both over the course of meiotic prophase I and between sexes.

## Sex-specific recombination-dependent regulation of SYP-2 and SYP-3 within the SC

During pachytene, one of the main functions of the SC is to facilitate and regulate recombination. In *C. elegans*, SPO-11-induced DSBs are formed in the context of fully assembled SC in early pachytene, and these breaks get repaired as the nuclei traverse through pachytene (*Gartner and Engebrecht, 2022*). By the transition to late pachytene, crossover recombination events are designated and marked by the pro-crossover protein COSA-1 (*Yokoo et al., 2012*). Notably, in *C. elegans* oocytes, the SC is modified in response to recombination and alterations of the SC influences and/or impairs recombination (*Almanzar et al., 2023*; *Cahoon et al., 2019*; *Colaiácovo et al., 2003*; *Gao et al., 2016*; *Gordon et al., 2021*; *Köhler et al., 2020*; *Libuda et al., 2013*; *Nadarajan et al., 2016*; *Rog et al., 2017*; *Sato-Carlton et al., 2018*; *Schild-Prüfert et al., 2011*). To determine whether the changes we observed in SYP-2 and SYP-3 accumulation during pachytene were caused by recombination, we assessed how the absence of recombination influenced the incorporation of each protein within the SC. To achieve this, we inhibited the formation of crossovers at different stages of recombination using two mutants: (1) *spo-11(me44)* mutants, which cannot form DSBs; and (2) *cosa-1(tm3298)* mutants, which cannot designate DSBs for crossover formation.

The amount of both GFP::SYP-2 and mCherry::SYP-3 loaded into the SC was significantly increased in *spo-11* mutant oocytes, but not spermatocytes (*Figure 3A and C*, *Figure 3—figure supplement 1*). The pattern of SYP-2 accumulation was altered in *spo-11* oocytes, in which SYP-2 amounts continued to increase within the SC displaying a 1.27-fold increase in SYP-2 amounts during early pachytene, 1.62-fold increase during mid pachytene, and 1.57-fold increase during late pachytene (*Figure 3A*). While the overall pattern of SYP-3 accumulation in *spo-11* oocytes remains very similar to wild-type with a progressive accumulation throughout pachytene, the total amounts of SYP-3 significantly increase within the SC (*Figure 3C*). In contrast, *spo-11* spermatocytes displayed mild changes in amounts of SYP-2 and SYP-3 in the SC compared to wild-type and the trend of SYP incorporation through pachytene was unaltered (*Figure 3A and C*). Additionally, western blot whole worm analysis of GFP::SYP-2 protein showed a slight, but not statistically significant, increase in SYP-2 protein levels in only oocytes (*Figure 3—figure supplement 2*). Thus, oocytes and spermatocytes differentially regulate the amount of SYP-2 protein in the absence of DSBs. Taken together, these results demonstrate that the incorporation of SYPs is regulated in a recombination-dependent and sex-specific manner during prophase I.

This sex-specific regulation of the SYPs in response to DSB formation was also observed in the crossover-deficient *cosa-1* mutants, albeit to a different degree in comparison to *spo-11* mutant oocytes (*Figure 3B and D*, *Figure 3—figure supplement 1*). Specifically, *cosa-1* oocytes did not

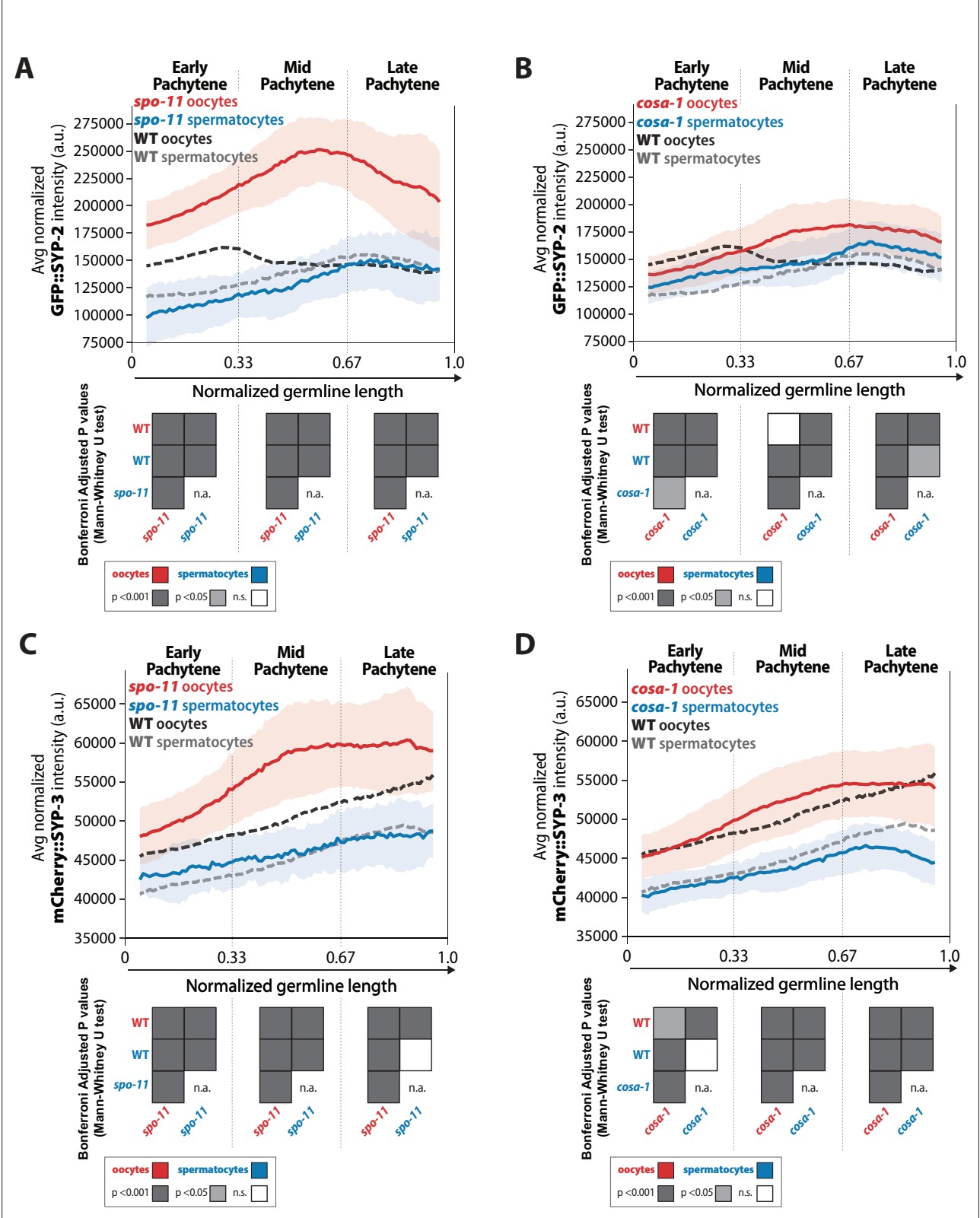

**Figure 3.** Recombination influences the incorporation of SYP-2 and SYP-3 in the synaptonemal complex (SC) differently in each sex. (**A, B**) Quantification of the mean intensity of GFP::SYP-2 per nucleus normalized by the volume of each nucleus (see 'Methods') throughout pachytene for *spo-11* (**A**) and *cosa-1* (**B**). (**C, D**) Quantification of the mean intensity of mCherry::SYP-3 per nucleus normalized by the volume of each nucleus (see 'Methods') throughout pachytene for *spo-11* (**C**) and *cosa-1* (**D**). Oocytes are shown in red with the standard deviation shown as a pale red band and spermatocytes

*Figure 3 continued on next page*

*Figure 3 continued*

are shown in blue with the standard deviation shown as a pale blue band. The mean intensity of GFP::SYP-2 (**A, B**) or mCherry::SYP-3 (**C, D**) for wild-type (WT) oocytes (black) and WT spermatocytes (gray) are shown as dashed lines. Heat maps below each pachytene region show the Bonferroni adjusted p-values from Mann–Whitney *U* tests, with dark gray indicating p<0.001, light gray indicating p<0.05, and white indicating not significant (n.s.). The self-comparison between spermatocyte *spo-11* or *cosa-1* mutants was not determined (n.a.). n values for the number of germlines and nuclei can be found in *Figure 3—source data 2*.

The online version of this article includes the following source data and figure supplement(s) for figure 3:

**Source data 1.** Raw sum intensity and normalized sum intensity per nucleus for GFP::SYP-2 and mCherry::SYP-3 in both sexes and all genotypes analyzed in *Figure 3*.

**Source data 2.** SC intensity n values for nuclei and germlines scored.

**Figure supplement 1.** Representative images of *spo-11* and *cosa-1* germlines with GFP::SYP-2 (**A**) and mCherry::SYP-3 (**B**) in oocytes and spermatocytes.

**Figure supplement 2.** Western blot analysis of SYP-2 protein levels in oocytes and spermatocytes.

**Figure supplement 2—source data 1.** Original western blot images, raw SYP-2 band intensity measurements, and normalized SYP-2 amounts in both sexes and all genotypes analyzed in this manuscript.

**Figure supplement 3.** GFP::SYP-2 aggregates in *cosa-1* at 25°C during mid and late pachytene.

increase the levels of SYP-2 and SYP-3 to the same amounts as observed in *spo-11* oocytes (*Figure 3*). In early pachytene, *cosa-1* oocytes showed a 0.8-fold decrease in SYP-2 amounts that changed in mid and late pachytene with SYP-2 amounts increasing by 1.16-fold and 1.22-fold over wild-type amounts, respectively (*Figure 3B*). Whereas *spo-11* oocytes increased the amount of assembled SYP-2 to a greater degree than *cosa-1* oocytes. Thus, SYP-2 and SYP-3 levels in oocytes can be differentially regulated depending on the specific recombination stage that is impeded. Similar to *spo-11* mutant spermatocytes, *cosa-1* mutant spermatocytes did not largely alter SYP-2 and SYP-3 levels (*Figure 3*). Even when recombination is hindered, the pattern of SYP-2 and SYP-3 incorporation throughout pachytene is not the same for each sex or between the SYPs. Overall, the SYPs within the SC of oocytes largely respond to alterations in recombination whereas the SC of spermatocytes do not significantly respond to recombination.

## Reduced SYP-2 causes altered SC assembly in oocytes

The SYP-specific changes in SC composition and in response to recombination defects raised the possibility that SYP protein dosage may regulate specific steps of recombination. To alter the dosage of each SYP, we used heterozygous null mutants for either *syp-2(ok307)* or *syp-3(ok785)* referred to as *syp-2/+* or *syp-3/+*. In oocytes, reducing the dosage of SYP-1, SYP-2, or SYP-3 by 60–70% was sufficient to permit SC assembly and crossover designation (*Libuda et al., 2013*). It remained unclear, however, whether altering the dosage of the SYPs also influenced chromosome pairing or the timing of SC assembly, which can also influence downstream meiotic processes like recombination (*Couteau et al., 2004*; *Couteau and Zetka, 2005*; *Goodyer et al., 2008*; *MacQueen et al., 2005*; *Martinez-Perez and Villeneuve, 2005*; *Mlynarczyk-Evans and Villeneuve, 2017*; *Nabeshima et al., 2004*; *Zhang et al., 2012*).

In *syp-2/+* and *syp-3/+* mutants, the transition zone (determined by DAPI-stained DNA morphology) was not significantly extended in either sex, indicating that chromosome pairing is not impaired by SYP dosage (*Figure 4—figure supplement 1*). Since SYP-1, SYP-2, and SYP-3 are dependent on each other for assembly, we assessed SC assembly using SYP-1 staining in *syp-2/+* and *syp-3/+* oocytes and spermatocytes (*Colaiácovo et al., 2003*; *MacQueen et al., 2002*; *Smolikov et al., 2007b*). SC assembly and/or the maintenance of full-length SC were altered by reducing the dosage of SYP-2 only in oocytes (*Figure 4*). In early pachytene, *syp-2/+* oocytes displayed more discontinuities in the SC along the chromosomes than both wild type and *syp-3/+*, suggesting that there is an SC assembly or maintenance defect in *syp-2/+* (*Figure 4A*, yellow arrowheads). Notably, these SC defects in early pachytene caused a significant increase in the length of the SC assembly zone in *syp-2/+* oocytes, but these mutants did maintain full-length SC after this zone (*Figure 4C and E*; p<0.001, Bonferroni p adjusted, Mann–Whitney). In contrast, *syp-2/+* and *syp-3/+* spermatocytes did not display any significant defects in SC assembly in early pachytene (*Figure 4B and D*). Additionally, *syp-2/+* or *syp-3/+* mutants in both sexes did not display any changes in SC disassembly or pachytene length, as indicated

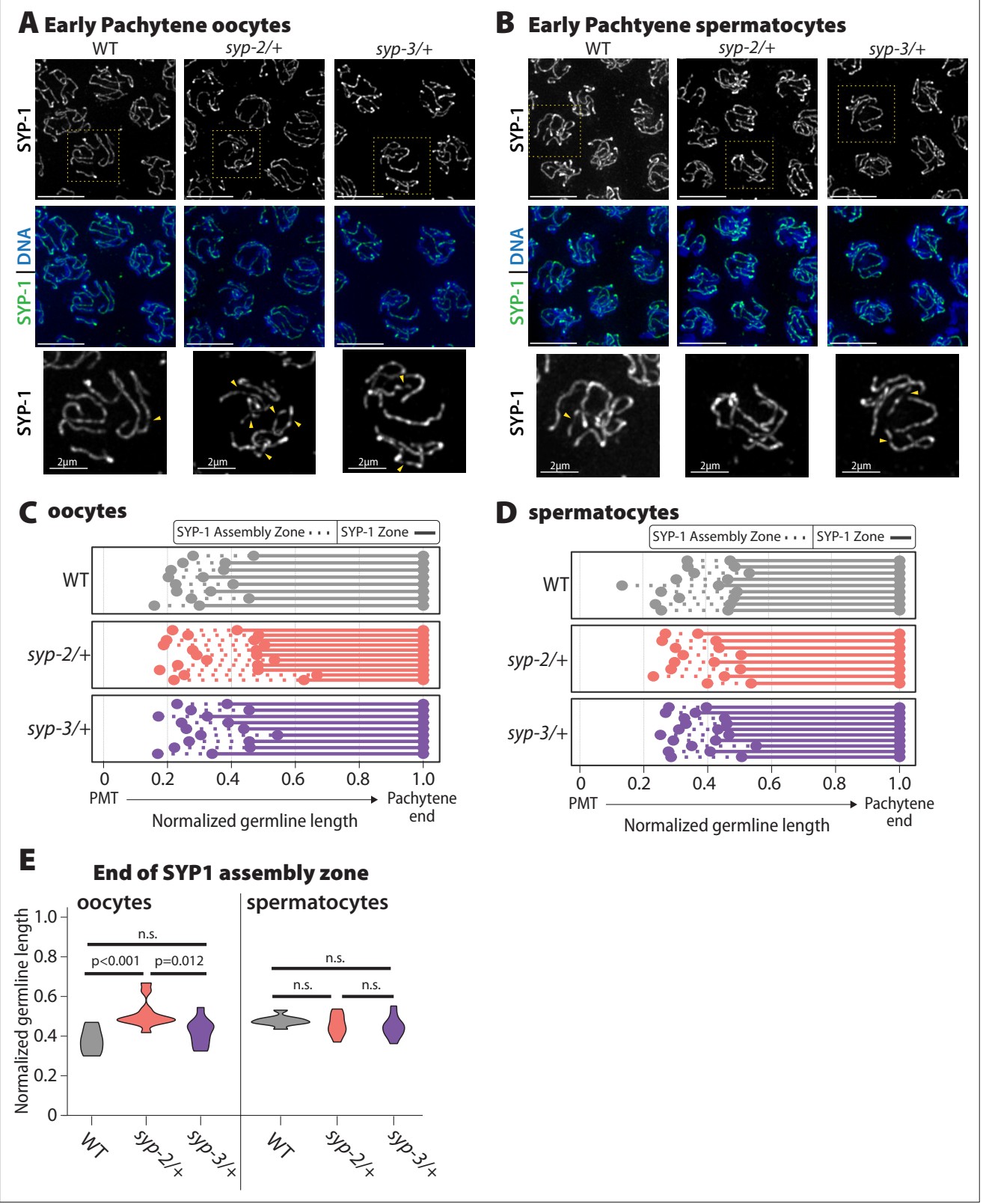

**Figure 4.** Oocyte SYP-1 assembly is uniquely sensitive to SYP-2 dosage. (**A, B**) Representative images early pachytene stained for SYP-1 in oocytes (**A**) and spermatocytes (**B**) from wild-type (WT), *syp-2/+*, and *syp-3/+*. Yellow dashed box shows the nucleus that is enlarged below each merged image. Yellow arrowheads indicate the regions where the SYP-1 signal is broken/not continuous. Scale bar represents 5 µm (**C, D**) Measurement of the relative length of the SYP-1 assembly zone in oocytes (**C**) and spermatocytes (**D**) from the pre-meiotic tip (PMT) to the end of pachytene in WT (gray), *syp-2/+*

*Figure 4 continued on next page*

*Figure 4 continued*

(pink), and *syp-3/+* (purple). Dashed lines represent the SC assembly zone and solid lines represent fully assembled synaptonemal complex (SC). (**E**) Quantification of the end of the SYP1 assembly zone in WT (gray), *syp-2/+* (pink), and *syp-3/+* (purple) in oocytes (left) and spermatocytes (right). All statistics are Bonferroni adjusted p values from Mann–Whitney *U* tests (n.s. = not significant). Oocyte data is from 8 WT germlines, 10 *syp-2/+* germlines, and 9 *syp-3/+* germlines. Spermatocyte data is from 9 WT germlines, 8 *syp-2/+* germlines, and 10 *syp-3/+* germlines.

The online version of this article includes the following source data and figure supplement(s) for figure 4:

**Source data 1.** Distance data from germlines where the length of SYP-1 assembly was determined (*Figure 4C–E*).

**Figure supplement 1.** Transition zone length and SUN-1 phosphorylation zone are unaltered by SYP dosage.

**Figure supplement 1—source data 1.** Distance data for the length of meiosis stages and SUN-1 S8P.

by DAPI morphology (*Figure 4*, *Figure 4—figure supplement 1*). We also checked SUN-1 S8 phosphorylation in both *syp-2/+* and *syp-3/+* to assess whether synapsis checkpoints were activated, but there was no significant change in the length of the SUN-1 S8 phosphorylation zone in either mutant or sex (*Figure 4—figure supplement 1*; *Woglar et al., 2013*). Thus, SC assembly and/or maintenance only in oocytes are sensitive to SYP-2 dosage during the early stages of pachytene and this defect is not severe enough to trigger synapsis checkpoint activation.

## SYP-2 and SYP-3 dosage regulate recombination via separate, sex-specific mechanisms

Since altering SYP dosage permitted assembly of full-length SC and did not impair chromosome pairing, we next assessed how reducing the dosage of SYP-2 and SYP-3 influenced recombination. To assess the mechanics of recombination, we used immunofluorescence to quantify and detect specific proteins that mark sites undergoing three different stages of recombination: (1) DSB formation with RAD-51 (*Colaiácovo et al., 2003*) (note: the subsequent removal of RAD-51 also indicates progression of a DSB down a repair pathway); (2) joint molecule formation with GFP::MSH-5 *Janisiw et al., 2018*; and (3) crossover designation with GFP::COSA-1 in oocytes (*Yokoo et al., 2012*) and OLLAS::COSA-1 in spermatocytes (*Janisiw et al., 2018*; see 'Methods'). For simplicity, here we refer to the tagged versions of GFP::MSH-5 and GFP::COSA-1 or OLLAS::COSA-1 as only the protein name either MSH-5 or COSA-1. The average number of foci of each protein was determined by using a sliding window of 0.01 along the normalized germline length, which was divided into early, mid, and late pachytene (see 'Methods).

### RAD-51 foci are relatively unaffected by SYP dosage

Similar to previous studies, we found that the number and timing of RAD-51 foci per nucleus during pachytene are very different between the sexes (*Figure 5A and B*, *Figure 5—figure supplement 1*; *Checchi et al., 2014*; *Jaramillo-Lambert and Engebrecht, 2010*). Oocytes initiate DSBs later in the germline than spermatocytes; however, both sexes progressively repair these breaks throughout pachytene (*Checchi et al., 2014*; *Colaiácovo et al., 2003*; *Jaramillo-Lambert and Engebrecht, 2010*; *Toraason et al., 2021*). Based on the RAD-51 foci counts, altering SYP dosage did not appear to have large effects on DSB initiation or subsequent progression through a repair pathway in either sex (*Figure 5A and B*, *Figure 5—figure supplement 1*). We did note changes in the number of RAD-51 foci specifically in oocytes of *syp-2/+* and *syp-3/+* during early pachytene (p<0.001; *Figure 5A*), but these slight alternations did not change the DSB repair dynamics with foci numbers declining at similar rates to wild-type during pachytene progression. Additionally, we checked DSB-2 staining in oocytes, which marks the region of the germline where DSBs are induced and found no significant changes in the DSB-2 zone in either mutant (*Figure 5—figure supplement 2*). Thus, DSB formation and repair are not largely impacted by altering the dosage of SYP-2 or SYP-3.

### SYP dosage influences MSH-5 foci via sex-specific mechanisms

Consistent with the early loading of RAD-51 in spermatocytes, we found that MSH-5 is also loaded earlier in spermatocytes than oocytes during pachytene (*Figure 5C and D*, *Figure 5—figure supplement 1*). Wild-type spermatocytes reach peak amounts of MSH-5 foci around the transition between early and mid pachytene, whereas oocytes have peak amounts of MSH-5 foci in mid pachytene. Moreover, the peak amount of MSH-5 foci loaded per nucleus is higher in spermatocytes than oocytes

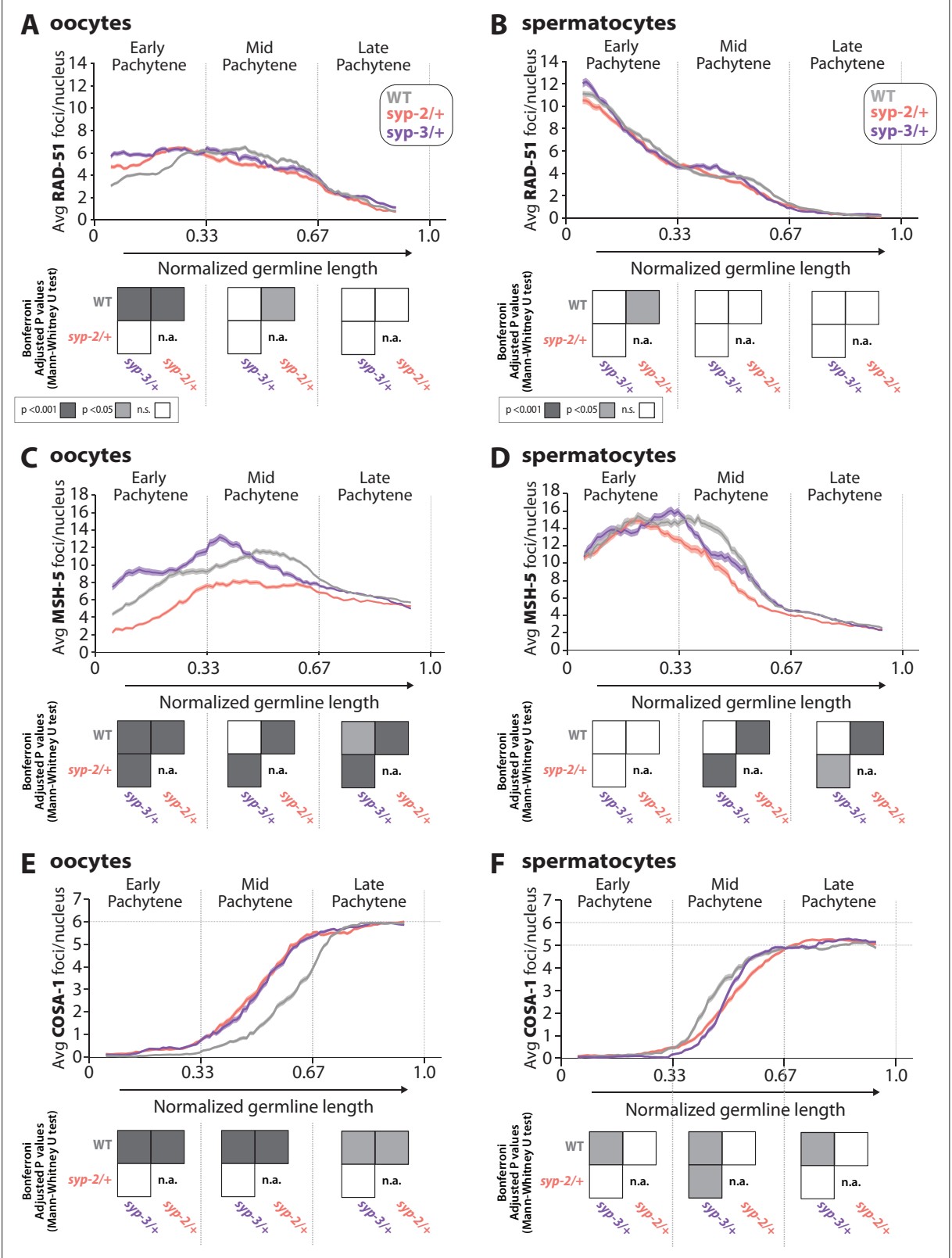

**Figure 5.** Sex-specific regulation of recombination by SYP-2 and SYP-3 dosage. (**A, B**) Quantification of the average number of RAD-51 foci per nucleus throughout pachytene in wild-type (WT, gray), *syp-2/+* (pink), and *syp-3/+* (purple) from oocytes (**A**) and spermatocytes (**B**). (**C, D**) Quantification of the average number of MSH-5 foci per nucleus throughout pachytene in WT (gray), *syp-2/+* (pink), and *syp-3/+* (purple) from oocytes (**C**) and spermatocytes (**D**). (**E, F**) Quantification of the average number of COSA-1 foci per nucleus throughout pachytene in WT (gray), *syp-2/+* (pink), and *syp-3/+* (purple)

*Figure 5 continued on next page*

*Figure 5 continued*

from oocytes (**E**) and spermatocytes (**F**). Heat maps below each pachytene region show the Bonferroni adjusted p-values from Mann–Whitney *U* tests, with dark gray indicating p<0.001, light gray indicating p<0.05, and white indicating not significant (n.s.). The self-comparison between *syp-2/+* was not determined (n.a.). n values for the number of germlines and nuclei can be found in *Figure 5—source data 2*.

The online version of this article includes the following source data and figure supplement(s) for figure 5:

**Source data 1.** RAD-51, MSH-5, COSA-1 foci per nucleus counts for wild-type, *syp-2/+* (syp2het), and *syp-3/+* (syp3het).

**Source data 2.** RAD-51, MSH-5, and COSA-1 n values for nuclei and germlines scored.

**Figure supplement 1.** Representative images of the oocytes (A) and spermatocytes (B) quantification in *Figure 5*.

**Figure supplement 2.** DSB-2 zone is unaltered in oocytes when SYP dosage is reduced.

**Figure supplement 2—source data 1.** Distance data from germlines where the length of DSB-2 zone was determined.

(mean MSH-5 oocytes 11.6 foci ± SD 6.0 vs. spermatocytes 15.4 ± SD 7.0). These differences in MSH-5 foci between the sexes contribute to a growing body of work, illustrating that the processing of recombination events is sexually dimorphic (*Brick et al., 2018*; *Checchi et al., 2014*; *Li et al., 2020*).

Reducing the dosage of SYP-2 and SYP-3 caused significant changes in the number of MSH-5 foci per nucleus during pachytene (*Figure 5C and D*, *Figure 5—figure supplement 1*), particularly in oocytes. *syp-2/+* oocytes showed significant reductions in the average number of MSH-5 foci throughout pachytene, indicating that the amount of SYP-2 is crucial to load and/or maintain MSH-5 at a DSB (*Figure 5C*; p<0.001, Bonferroni adjusted p, Mann–Whitney). *syp-3/+* oocytes displayed significant increases in the average number of MSH-5 foci per nucleus during early and mid pachytene (*Figure 5C*, p<0.001, Bonferroni adjusted p, Mann–Whitney). However, during mid pachytene, *syp-3/+* oocytes rapidly lost MSH-5 foci earlier than wild-type. Taken together, our data suggests that SYP-3 dosage regulates the timing of MSH-5 loading and off-loading (or maintenance at a DSB) in oocytes.

In spermatocytes, only reducing the dosage of SYP-2 caused significant changes in MSH-5 foci during pachytene (*Figure 5D*, *Figure 5—figure supplement 1*). *syp-2/+* spermatocytes initially form MSH-5 foci to levels similar to wild-type in early pachytene. However, during mid and late pachytene *syp-2/+* spermatocytes rapidly lose MSH-5 foci, suggesting that SYP dosage is important for the maintenance of MSH-5 at joint molecules (*Figure 5D*; p<0.001, Bonferroni adjusted p, Mann–Whitney). Notably, for both sexes the dosage of SYP-2 appears to be important for MSH-5 stability, suggesting a conserved role for SYP-2 in both sexes. Additionally, unlike oocytes, SYP-3 dosage does not change MSH-5 foci during pachytene in spermatocytes, illustrating a sex-specific role for SYP-3 in regulating MSH-5 at DSB sites during recombination.

## SYP dosage alters the timing of COSA-1 foci loading during pachytene

Unlike the sexually dimorphic DNA repair dynamics observed in RAD-51 and MSH-5, both oocytes and spermatocytes load COSA-1 in mid pachytene, and by late pachytene, all 6 COSA-1 foci in oocytes and 5–6 COSA-1 foci in spermatocytes have been established (*Figure 5E and F*, *Figure 5—figure supplement 1*; *Cahoon et al., 2023*; *Yokoo et al., 2012*). However, altering the dosage of SYP-2 and SYP-3 causes significant changes in the timing of COSA-1 loading in both oocytes and spermatocytes (*Figure 5E and F*, *Figure 5—figure supplement 1*). In both *syp-2/+* and *syp-3/+* oocytes, the loading of COSA-1 foci was shifted to occur earlier in mid pachytene than wild-type (*Figure 5E*; *syp-2/+* p<0.001, *syp-3/+* p<0.001, Bonferroni adjusted p, Mann–Whitney). In contrast, *syp-3/+* spermatocytes exhibit a delay in the loading of COSA-1 foci during mid pachytene (*Figure 5F*; p<0.05, Bonferroni adjusted p, Mann–Whitney). Although *syp-2/+* spermatocytes also displayed a potential delay in the loading of COSA-1, the difference is not statistically different from wild-type (*Figure 5F*; p=1, Bonferroni adjusted p, Mann–Whitney). Taken together, dosage of SYP-2 and SYP-3 regulates the sex-specific timing of crossover designation.

By late pachytene, the final number of COSA-1 foci in oocytes (six foci) and spermatocytes (five or more foci) was not changed from the required one crossover per chromosome, thereby suggesting that SYP dosage does not largely influence the ability of each sex to designate the crossovers on all chromosomes (*Figure 5E and F*, *Figure 5—figure supplement 1*). Further, in oocytes, we observed six DAPI staining bodies at diakinesis for both *syp-2/+* (30/30 oocytes with six DAPI bodies) and

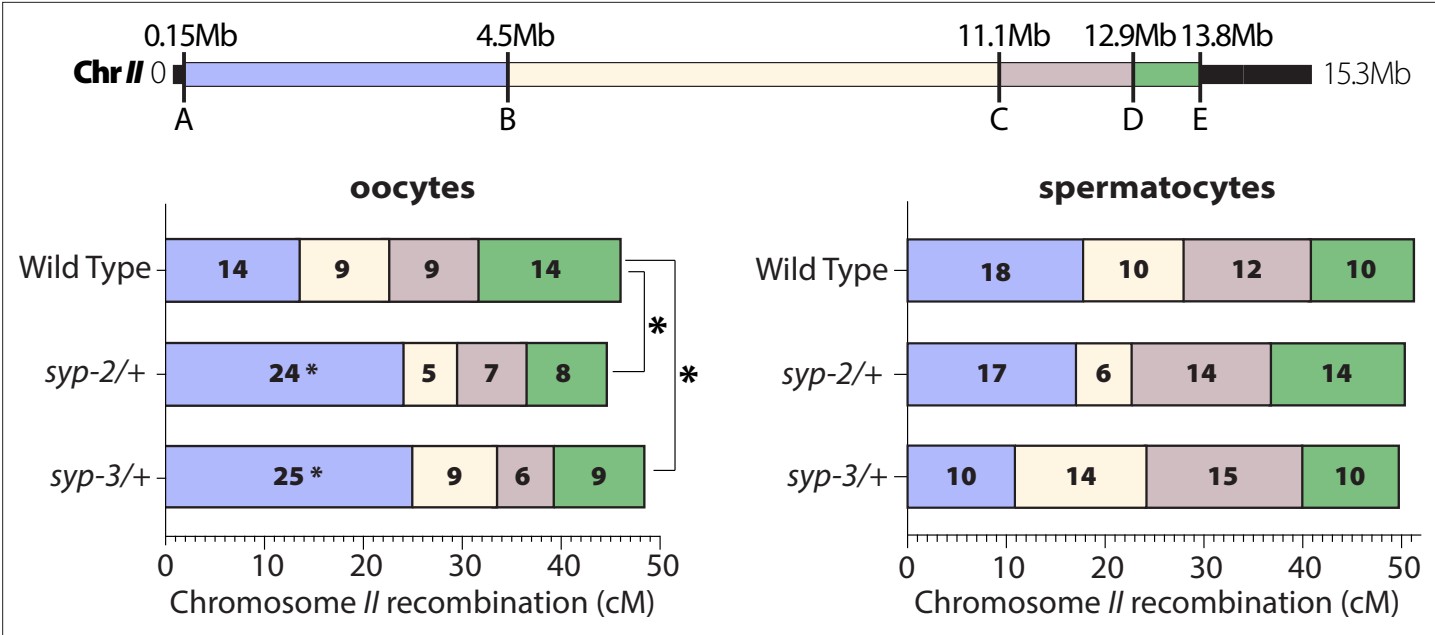

**Figure 6.** SYP dosage influences the crossover landscape in only oocytes. Recombination SNP mapping of chromosome *II* in WT, *syp-2/+*, and *syp-3/+* from oocytes (left) and spermatocytes (right) (see 'Methods' for details). A diagram of the 15.3 Mb chromosome *II* shows the megabase location of each SNP assayed (A–E), and the colored boxes between each SNP show the intervals where crossovers were assessed. The map length (cM) is indicated in each crossover interval. The asterisks next to the map lengths indicate significance based on Fisher's exact tests compared between the mutants and wild-type (*syp2/+* p=0.0449; *syp3/+* p=0.0170). The asterisks outside the bars indicate significance based on chi-squared tests between mutants and wild-type (*syp2/+* p=0.0343; *syp3/+* p=0.0333). The worm counts for these plots can be found in *Table 1*.

The online version of this article includes the following source data and figure supplement(s) for figure 6:

**Figure supplement 1.** *X* chromosome may have some chromosome distortion in *syp-2/+* oocytes.

**Figure supplement 1—source data 1.** Worm counts for chromosome *X* SNP mapping recombination.

*syp-3/+* (30/30 oocytes with six DAPI bodies), indicating that the six COSA-1 foci per nucleus at late pachytene mature into six crossovers that link the homologous chromosomes together at diakinesis.

## SYP-dosage regulates crossover landscape

Since we found that manipulating the dosage of SYP-2 and SYP-3 altered multiple steps of recombination and previous studies indicate that the SC can regulate crossover numbers (*Colaiácovo et al., 2003*; *Gordon et al., 2021*; *Hayashi et al., 2010*; *Köhler et al., 2020*; *Libuda et al., 2013*; *MacQueen et al., 2002*; *Smolikov et al., 2007b*), we wanted to determine whether SYP-2 and SYP-3 dosage influences the recombination landscape by changing where crossovers are positioned along the length of the chromosome. To assess the recombination landscape, we used established single-nucleotide polymorphism (SNP) recombination mapping between two *C. elegans* isolates (Bristol and

**Table 1.** Chromosome *II* SNP mapping recombination.

| | | Recombinant Intervals | | | | | |
|---|---|---|---|---|---|---|---|
| **Sex** | **Genotype** | **A—B** | **B—C** | **C—D** | **D—E** | **Non-recombinant** | **Total worms** |
| | Wild-type | 26 | 17 | 17 | 27 | 103 | 190 |
| | *syp-2/+* | 45 | 10 | 13 | 16 | 102 | 186 |
| Oocytes | *syp-3/+* | 44 | 15 | 10 | 16 | 90 | 175 |
| | Wild-type | 31 | 17 | 21 | 17 | 79 | 165 |
| | *syp-2/+* | 29 | 10 | 24 | 23 | 84 | 170 |
| Spermatocytes | *syp-3/+* | 20 | 25 | 29 | 18 | 92 | 184 |

Hawaiian) to identify crossovers (see 'Methods'; *Bazan and Hillers, 2011*). For oocytes, we mapped recombination on both chromosome *II* and the *X* chromosome. Since male worms have an unpaired *X* chromosome and do not form crossovers on this sex chromosome, only recombination on chromosome *II* was mapped in spermatocytes.

On chromosome *II*, both *syp-2/+* and *syp-3/+* only altered the crossover landscape in oocytes (*Figure 6*, *Table 1*). In spermatocytes, both *syp-2/+* and *syp-3/+* displayed no significant changes in crossover frequencies across all of chromosome *II*. In contrast, both *syp-2/+* and *syp-3/+* oocytes displayed significant changes in the crossing over distribution on chromosome *II* (*Figure 6*; *syp2/+* p=0.0343, *syp3/+* p=0.0333, chi-squared). Specifically, both mutants increased crossing over in the first interval (A to B) on the left side of chromosome *II* by ~10 cM each (*Figure 6*; wild-type 14 cM, *syp-2/+* 24 cM, *syp-3/+* 25 cM; *syp2/+* p=0.0449, *syp3/+* p=0.0170, Fisher's exact test). Intriguingly, this first interval is also where the pairing center is located on chromosome *II*. Thus, one possible explanation for the elevated crossing over in the interval is that by reducing the SYP dosage crossovers are now more often formed where the SC is first assembled at the pairing center (*Hayashi et al., 2010*; *Rog and Dernburg, 2015*).

On the *X* chromosome, *syp-2/+* oocytes displayed a significant decrease in crossover frequency along the entire chromosome (*Figure 6—figure supplement 1*; wild-type 51 cM, *syp-2/+* 32 cM; p=0.0343, chi-squared). This decrease was not observed in *syp-3/+* oocytes, which showed no significant changes in crossover frequency to wild-type for *X* chromosome recombination (*Figure 6—figure supplement 1*; wild-type 51 cM, *syp-3/+* 45 cM; p=0.5849, chi-squared). The large decrease in crossing over in *syp-2/+* suggests that these mutants should have a significant amount of *X* chromosome nondisjunction or missegregation. In *C. elegans*, mutants with frequent *X* chromosome nondisjunction produce a high incidence of male (Him) phenotype as male worms are hemizygous for the *X* chromosome. Notably, *syp-2/+* did not display an elevation in the frequency of male progeny nor did they have an increase in dead eggs (*Figure 6—figure supplement 1*; wild-type 0.4% dead eggs and 0.2% Him, *syp-2/+* 0.4% dead eggs and 0.2% Him). Since the SNP recombination mapping experiment is performed in a hybrid Bristol/Hawaiian background, we also checked the hybrid background for male progeny and dead eggs. Due to known meiotic drive elements between these strains, the Bristol/Hawaiian hybrids produce more dead eggs (*Seidel et al., 2011*; *Seidel et al., 2008*). However, *syp-2/+* Bristol/Hawaiian hybrids did not display a higher incidence of dead eggs or more male progeny than the wild-type Bristol/Hawaiian hybrid (*Figure 6—figure supplement 1*; wild-type 19.4% dead eggs and 0.2% Him; *syp-2/+* 23.3% dead eggs and 0.1% Him). Thus, it remains unclear as to why reducing SYP-2 causes a significant decrease in recombination without a corresponding increase in *X* chromosome nondisjunction. One possible explanation is that *syp-2/+* have a chromosome distortion event occurring in the later stages of meiosis that is removing recombinant *X* chromosomes. Future studies are needed to understand what is happening to the recombinant *X* chromosomes in *syp-2/+*.

In comparison to wild-type, the broods from *syp-3/+* hermaphrodites exhibited both an increased lethality (more dead eggs; *Figure 6—figure supplement 1*, wild-type Bristol 0.4% and hybrid 19.4% dead eggs; *syp-3/+* Bristol 1.2% and hybrid 27.0% dead eggs) as well as a higher incidence of male progeny than wild-type (*Figure 6—figure supplement 1*, wild-type Bristol 0.2% and hybrid 0.2% HIM; *syp-3/+* Bristol 0.7% and hybrid 1.6% HIM, Bristol p<0.0001, hybrid P<0.0001, chi-squared). In addition, *syp-3/+* also produced a higher rate of progeny with dumpy and/or uncoordinated mutant phenotypes than both wild-type and *syp-2/+* (*Figure 6—figure supplement 1*, Bristol 0.06% mutant progeny, *syp-2/+* Bristol 0.06% mutant progeny, *syp-3/+* Bristol 0.47% mutant progeny; wild-type Bristol p=0.0207, *syp-2/+* Bristol p=0.0388, Fisher's exact test). SYP proteins have been implicated in regulating DSB repair pathway choice by preventing access to error-prone and sister chromatid repair (*Láscarez-Lagunas et al., 2022*; *Lemmens et al., 2013*; *Macaisne et al., 2018*; *Rosu et al., 2011*; *Smolikov et al., 2007a*; *Yin and Smolikove, 2013*). Additionally, SYP-3 directly interacts with the pro-crossover protein BRC-1 to promote crossover DSB repair (*Janisiw et al., 2018*). Thus, the mutant phenotypes observed in progeny from *syp-3/+* hermaphrodites are likely due to error-prone repair using nonhomologous end joining, which is used by *syp-3* nulls to repair persisting DSBs (*Smolikov et al., 2007b*). These results suggest that SYP-3 amounts are important for suppressing error-prone repair. Notably, in contrast to oocytes, spermatocyte fertility was unaffected in *syp-2/+* or *syp-3/+* (*Figure 6—figure supplement 1*). Thus, reducing the dosage of SYP-2 or SYP-3 has sexually dimorphic consequences on fertility.

## SYP-2 and SYP-3 dosage influences the SC composition during pachytene

The SYP dosage-dependent regulation of recombination (*Figures 4 and 5*), different patterns of SYP protein incorporation throughout pachytene (*Figure 2*), and independent regulation of SYP loading into the SC (*Figures 2 and 3*) suggest that SYP accumulation in the SC may be dynamically altered during pachytene to regulate the steps of recombination. Thus, reducing the gene dosage of each SYP may influence how the SYPs are accumulating within the SC, which could influence the regulation of recombination. To determine how *syp* gene dosage influences the incorporation of each SYP during pachytene, we used heterozygous null mutants of each *syp* gene with a wild-type fluorescently tagged GFP::SYP-2 or mCherry::SYP-3: *syp-2(ok307)/gfp::syp-2* (referred to as *syp-2/+*) or *syp-3(ok785)/mCherry::syp-3* (referred to as *syp-3/+*). Indeed, we found that the dosage of *syp-2* and *syp-3* influenced both SYP-2 and SYP-3 accumulation during pachytene in both sexes (*Figure 7*).

Altering the *syp-2* gene dosage resulted in both sexes to initially have an ~0.7-fold decrease in SYP-2 amounts within the SC; however, SYP-2 levels progressively increased throughout pachytene until wild-type amounts were reached or pachytene ended (*Figure 7A*). Western blot analysis revealed that GFP::SYP-2 protein levels were not significantly increased in *syp-2/+* oocytes and spermatocytes (*Figure 3—figure supplement 2*). Overall, these results suggest that both sexes attempt to compensate for the haploinsufficiency of losing a *syp-2* gene by loading as much SYP-2 into the SC as possible during pachytene.

In contrast, SYP-3 accumulation in *syp-2/+* mutants displayed sex-specific responses (*Figure 7B*). In oocytes, reducing the *syp-2* gene dosage resulted in a slight decrease in SYP-3 levels in early pachytene that becomes more pronounced in late pachytene, which we noted was the same window when SYP-2 protein levels reached wild-type amounts in *syp-2/+* mutants (*Figure 7B*; $p<0.001$, Bonferroni adjusted, Mann–Whitney). In spermatocytes, reducing the *syp-2* gene dosage resulted in an increase in SYP-3 amount in early pachytene that are indistinguishable from wild-type by late pachytene (*Figure 7B*). Thus, while altering *syp-2* gene dosage caused both sexes to respond similarly with reduced loading of SYP-2 into assembled SC, SYP-3 levels within assembled SC were inversely affected via sex-specific mechanisms.

We found that altering *syp-3* gene dosage resulted in the same overall changes to SYP-2 and SYP-3 amounts in each sex as altering *syp-2* gene dosage, albeit the degree to which each SYP changed was different (*Figure 7*). Specifically, reducing *syp-3* gene dosage caused an initial reduction in SYP-2 levels in both sexes that was less than the reduction in *syp-2/+* (*Figure 7A and C*; early pachytene *syp-2/+* oocytes 0.72-fold change vs. *syp-3/+* oocytes 0.87-fold change; early pachytene *syp-2/+* spermatocytes 0.71-fold change vs. *syp-3/+* spermatocytes 0.89-fold change). However, similar to *syp-2/+*, both sexes progressively increase SYP-2 amounts in *syp-3/+* until wild-type levels were reached or pachytene ended (*Figure 7A and C*). Additionally, western blot analysis of GFP::SYP-2 protein levels showed that both sexes only slightly increased SYP-2 protein levels (*Figure 3—figure supplement 2*). Thus, both sexes respond to alterations in *syp* gene dosages with similar trends of SYP-2 accumulation within the SC.

In oocytes, reducing *syp-3* gene dosage caused a stronger decrease in overall SYP-3 protein amounts compared to *syp-2/+* (*Figure 7B and D*, early pachytene *syp-2/+* 0.97-fold change vs. *syp-3/+* 0.91-fold change, mid pachytene 0.99-fold change vs. *syp-3/+* 0.90-fold change, late pachytene 0.91-fold change vs. *syp-3/+* 0.85-fold change). In contrast, *syp-3/+* spermatocytes displayed an increase in SYP-3 levels that were similar to the increase in SYP-3 observed in *syp-2/+* spermatocytes (*Figure 7B and D*). Thus, the dosage of each SYP gene influences the accumulation of SYP-2 and SYP-3 within the assembled SC similarly for each sex. Oocytes respond to reduced *syp* gene dosage by decreasing both SYP-2 and SYP-3 within the SC, while spermatocytes respond to reduced *syp* gene dosage by decreasing SYP-2 and increasing SYP-3 in the SC. Overall, these differences are likely influencing and/or responding to the changes we observed in recombination for both *syp-2/+* and *syp-3/+* (*Figure 5*) and suggest that the dosage of each SYP throughout pachytene regulates recombination in a sex-specific manner (see 'Discussion').

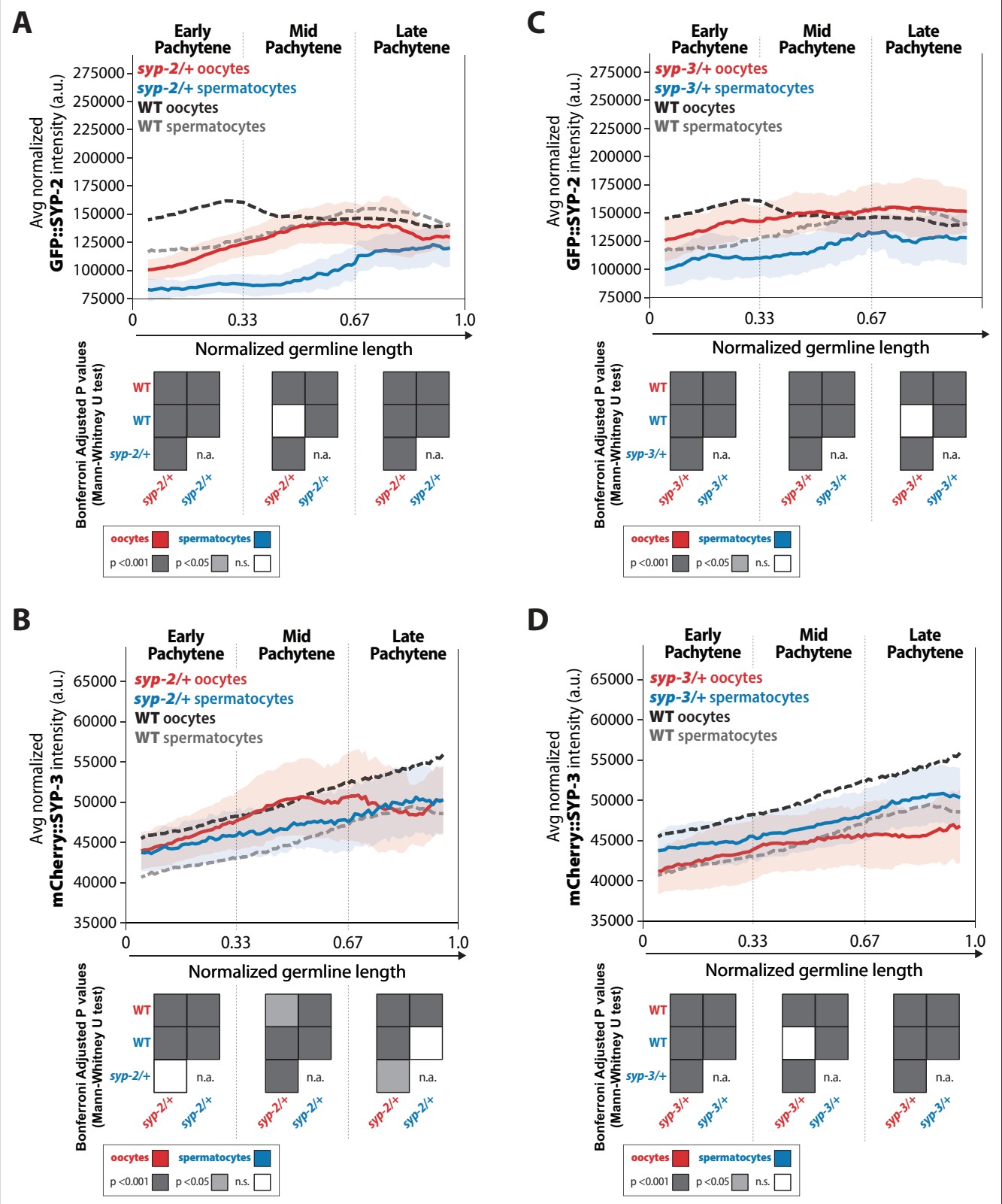

**Figure 7.** *syp-2* and *syp-3* gene dosage influence the amount of each SYP loaded within the synaptonemal complex (SC) via sex-specific mechanisms. (**A, B**) Quantification of the mean intensity of GFP::SYP-2 (**A**) and mCherry::SYP-3 (**B**) per nucleus normalized by the volume of each nucleus (see 'Methods') throughout pachytene for *syp-2/+*. (**C, D**) Quantification of the mean intensity of GFP::SYP-2 (**C**) mCherry::SYP-3 (**D**) per nucleus normalized by the volume of each nucleus throughout pachytene (normalized, see 'Methods') for *syp-3/+*. Oocytes are shown in red with the standard deviation

*Figure 7 continued on next page*

*Figure 7 continued*

shown as a pale red band, and spermatocytes are shown in blue with the standard deviation shown as a pale blue band. The mean intensity of GFP::SYP-2 (**A, B**) and mCherry::SYP-3 (**C, D**) for wild-type (WT) oocytes (black) and WT spermatocytes (gray) are shown as dashed lines. Heat maps below each pachytene region show the Bonferroni adjusted p-values from Mann–Whitney *U* tests, with dark gray indicating p<0.001, light gray indicating p<0.05, and white indicating not significant (n.s.). The self-comparison between spermatocyte *syp-2/+* or *syp-3/+* mutants was not determined (n.a.). n values for the number of germlines and nuclei can be found in *Figure 7—source data 2*.

The online version of this article includes the following source data for figure 7:

**Source data 1.** Raw sum intensity and normalized sum intensity per nucleus for GFP::SYP-2 and mCherry::SYP-3 in both sexes and all genotypes in *Figure 7*.

**Source data 2.** Synaptonemal complex (SC) intensity n values for nuclei and germlines scored.

## SYP dosage differentially influences SYP-5 and SYP-6 composition within the SC

The ability of both SYP-2 and SYP-3 to affect the incorporation of each other within the SC suggested that the dosage of each SYP may influence and/or regulate the incorporation of other SYP proteins in the SC. Since the inability to functionally tag SYP-1 and SYP-4 with fluorescent proteins precluded analysis of these two proteins, we assessed how the composition of SYP-5 and SYP-6 within the SC was influenced by *syp-2* and *syp-3* gene dosage in both oocytes and spermatocytes. Using endogenously CRISPR tagged lines containing either SYP-5::GFP or SYP-6::GFP (*Zhang et al., 2020*) combined with heterozygous null mutants of *syp-2* (*syp-2/+*) or *syp-3* (*syp-3/+*), we found that *syp-2* and *syp-3* gene dosage can influence SYP-5 and SYP-6 incorporation in the SC via sex-specific mechanisms (*Figure 8*).

### Sexually dimorphic SYP-5 accumulation is differentially influenced by syp-2 and syp-3 gene dosage

Similar to the SYP-2 and SYP-3, the composition of SYP-5 within the SC is also sexually dimorphic. In wild-type oocytes, SYP-5::GFP accumulation in the SC progressively increased throughout pachytene, matching previous observations of SYP-5 localization (*Figure 8A*; *Hurlock et al., 2020*; *Zhang et al., 2020*). While wild-type spermatocytes also displayed a progressive increase in SYP-5::GFP through pachytene, the total amount of SYP-5::GFP was significantly less than oocytes (*Figure 8A*, p<0.001, Bonferroni adjusted, Mann–Whitney). Notably, the sex-specific difference in wild-type SYP-5 amounts was especially apparent in the late pachytene region where oocytes have 1.8-fold more SYP-5 within the SC than spermatocytes. Intriguingly, the accumulation pattern of wild-type SYP-3 and SYP-5 was very similar, progressively increasing in protein levels loaded throughout pachytene and with spermatocytes loading overall less protein (*Figures 2 and 8*).

Examining SYP-5 accumulation in both *syp-2/+* and *syp-3/+* showed more similarities between SYP-5 and SYP-3 accumulation in oocytes. In *syp-2/+* and *syp-3/+* oocytes, SYP-5::GFP accumulation significantly decreased within the SC throughout pachytene, which was the same response SYP-3 levels had to altered *syp-2* and *syp-3* gene dosage (*Figures 8A, 7B and D*). In spermatocytes, *syp-2/+* did not influence SYP-5::GFP amounts, but *syp-3/+* did significantly decrease SYP-5::GFP accumulation through pachytene (*Figure 8A*). This spermatocyte SYP-5 result contrasts with spermatocyte SYP-3 where reduced *syp-2* and *syp-3* gene dosage caused increased SYP-3 levels in both cases (*Figure 7B and D*). Thus, in oocytes, reducing the dosage of either *syp-2* or *syp-3* causes a reduction in both SYP-5 and SYP-3 levels within the SC during pachytene, thereby suggesting that SYP-5 and SYP-3 may be regulated similarly with the oocyte SC. Whereas in spermatocytes, there may be differential regulation of SYP-5 and SYP-3, which potentially indicates different roles for these proteins during spermatocyte meiosis (see 'Discussion).

### Sexually dimorphic localization of SYP-6

SYP-6 is the paralog of SYP-5 and has some functional redundancy with SYP-5 in SC assembly and synapsis in oocytes (*Hurlock et al., 2020*; *Zhang et al., 2020*). While both SYP-5 and SYP-6 are assembled into the SC within the same region of the germline, SYP-5 and SYP-6 are disassembled at different times (*Hurlock et al., 2020*; *Zhang et al., 2020*). In oocytes, SYP-5 is disassembled with the other SYPs in diplotene, whereas SYP-6 is disassembled in mid pachytene and is largely absent from late pachytene nuclei (*Hurlock et al., 2020*; *Zhang et al., 2020*). Since SYP-6 localization in spermatocytes

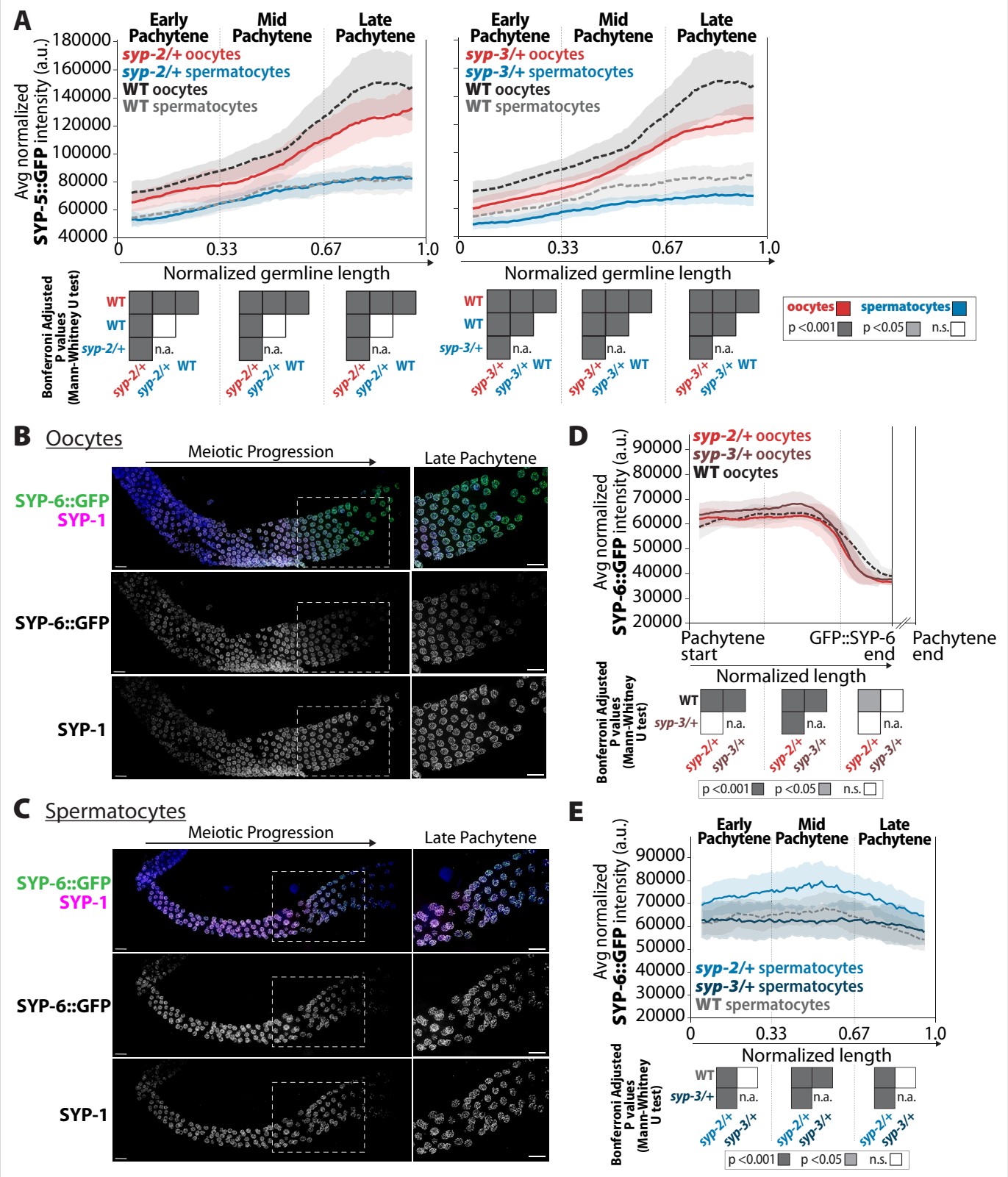

**Figure 8.** *syp-2* and *syp-3* gene dosage impacts the composition of SYP-5 and SYP-6 within the synaptonemal complex (SC). (**A**) Quantification of the mean intensity of SYP-5::GFP per nucleus normalized by the volume of each nucleus (see 'Methods') throughout pachytene for *syp-2/+* (left plot) and *syp-3/+* (right plot). Mutants are in solid lines with oocytes in red and spermatocytes in blue. Wild-type is in dashed lines with oocytes in black and spermatocytes in gray. The pale shading around each line is the standard deviation. (**B, C**) Representative images of wild-type germlines stained

*Figure 8 continued on next page*

*Figure 8 continued*

with SYP-6::GFP (green) and SYP-1 (magenta) in germlines with oocytes (**B**) and spermatocytes (**C**). The white dashed box shows the region enlarged in the image on the right. Scale bar represents 10 μm. (**D**) Quantification of the mean intensity of oocyte SYP-6::GFP per nucleus normalized by the volume of each nucleus throughout pachytene for *syp-2/+* (bright red) and *syp-3/+* (dark red). Since SYP-6 disassembles prior to the end of pachytene, the germline length is normalized to the germline region with SYP-6::GFP signal starting at the beginning of pachytene (Pachytene start) to the end of the SYP-6::GFP signal (SYP-6::GFP end) (see 'Methods'). The broken x-axis indicates the unknown distance to the end of pachytene (Pachytene end). Mutants are in solid lines, wild-type oocytes are in a dashed line, and the pale shading around each line represents the standard deviation. (**E**) Quantification of the mean intensity of spermatocyte SYP-6::GFP per nucleus normalized by the volume of each nucleus (see 'Methods') throughout pachytene for *syp-2/+* (bright blue) and *syp-3/+* (dark blue). Mutants are in solid lines, wild-type spermatocytes are in a dashed line, and the pale shading around each line represents the standard deviation. Heat maps below plots in panels (**A**), (**D**), and (**E**) show the Bonferroni adjusted p-values from Mann–Whitney *U* tests, with dark gray indicating p<0.001, light gray indicating p<0.05, and white indicating not significant (n.s.). The self-comparison between spermatocyte *syp-2/+* or *syp-3/+* mutants was not determined (n.a.). n values for the number of germlines and nuclei can be found in *Figure 8—source data 2*.

The online version of this article includes the following source data and figure supplement(s) for figure 8:

**Source data 1.** Raw sum intensity and normalized sum intensity per nucleus for SYP-5::GFP and SYP-6::GFP in both sexes and all genotypes in *Figure 8*.

**Source data 2.** Synaptonemal complex (SC) intensity n values for nuclei and germlines scored.

**Figure supplement 1.** Representative images of wild-type germlines stained with SYP-5::GFP and SYP-1 in oocytes and spermatocytes.

was unknown, we used immunofluorescence staining of SYP-1 and SYP-6::GFP to determine whether spermatocytes displayed the same early disassembly of SYP-6 (see 'Methods'). Surprisingly, unlike oocytes, SYP-6 persists in spermatocytes through late pachytene and disassembling in diplotene with SYP-1 with all the other SYPs (*Figure 8B and C*). SYP-5 localization was identical in each sex throughout pachytene, where it co-localized with SYP-1 (*Figure 8—figure supplement 1*; *Hurlock et al., 2020*; *Zhang et al., 2020*). Thus, the retention of SYP-6 in late pachytene SC indicates that the composition of late pachytene SC is sexually dimorphic.

Since reducing *syp-2* and *syp-3* gene dosage led to changes in SYP-5 levels within the SC, we assessed whether similar changes in SYP-6 levels occurred in *syp-2/+* and *syp-3/+* heterozygotes. Similar to *Zhang et al., 2020*, we also noticed in oocytes that SYP-6 has a different incorporation pattern than SYP-5 (*Figure 8D*). Since SYP-6 disassembles prior to the end of late pachytene in oocytes, we measured the intensity of SYP-6::GFP per nucleus from the start of pachytene until the end of the GFP::SYP-6 signal (*Figure 8*, see 'Methods'). In wild-type oocytes, SYP-6::GFP remained at a fairly constant amount before it was disassembled in mid pachytene. This SYP-6 result contrasts with SYP-5, which progressively increases throughout pachytene (*Figure 8A*). Further, unlike SYP-5, the *syp-2* and *syp-3* gene dosage does not appear to have a strong effect on altering the amount of SYP-6::GFP in oocytes (*Figure 8D*). Thus, SYP-5 and SYP-6 within oocyte SC are differentially regulated, thereby suggesting that these proteins might have some non-redundant roles during meiosis (*Zhang et al., 2020*).

Given that SYP-6 is retained to the end of pachytene in spermatocytes, we assessed the incorporation of SYP-6 throughout pachytene in spermatocytes (*Figure 8E*, see 'Methods'). In early pachytene, both wild-type oocytes and spermatocytes display similar amounts of SYP-6::GFP (*Figure 8D and E*). By mid pachytene, while oocytes are disassembling SYP-6, spermatocytes only display a slight reduction in SYP-6::GFP levels from mid/late pachytene to the end of pachytene (*Figure 8D and E*). Unlike oocyte SC, altering the dosage of SYP-2 influences SYP-6::GFP incorporation within the spermatocyte SC. Specifically, *syp-2/+* spermatocytes display a significant increase in the incorporation of SYP-6::GFP throughout pachytene (*Figure 8E*, p<0.001, Bonferroni adjusted, Mann–Whitney). In contrast, *syp-3/+* spermatocytes largely did not alter the incorporation of SYP-6::GFP during pachytene progression (*Figure 8E*).

Taken together, the regulation of SYP-2, SYP-3, SYP-5, and SYP-6 is different in both spermatocytes and oocytes, suggesting sex-specific roles for each protein during meiosis. Further, altering SYP-2 and SYP-3 dosage appears to cause global, sexually dimorphic changes within the SC that is likely a response to and/or consequence of the defects in specific steps of recombination.

## Discussion

### Sexually dimorphic regulation of recombination by the SC

One of the roles of the SC during meiosis is regulating recombination to promote the establishment of a crossover on each homolog pair. Our data here demonstrate that the SC in *C. elegans* regulates recombination via sexually dimorphic mechanisms. We show that the central region proteins SYP-2 and SYP-3 have sex-specific differences in protein turnover rates within the SC that may influence the sexually dimorphic composition of the SC. Specifically, SYP-2, SYP-3, SYP-5, and SYP-6 are incorporated into the SC at different levels via sex-specific mechanisms. Moreover, we find that SYP protein

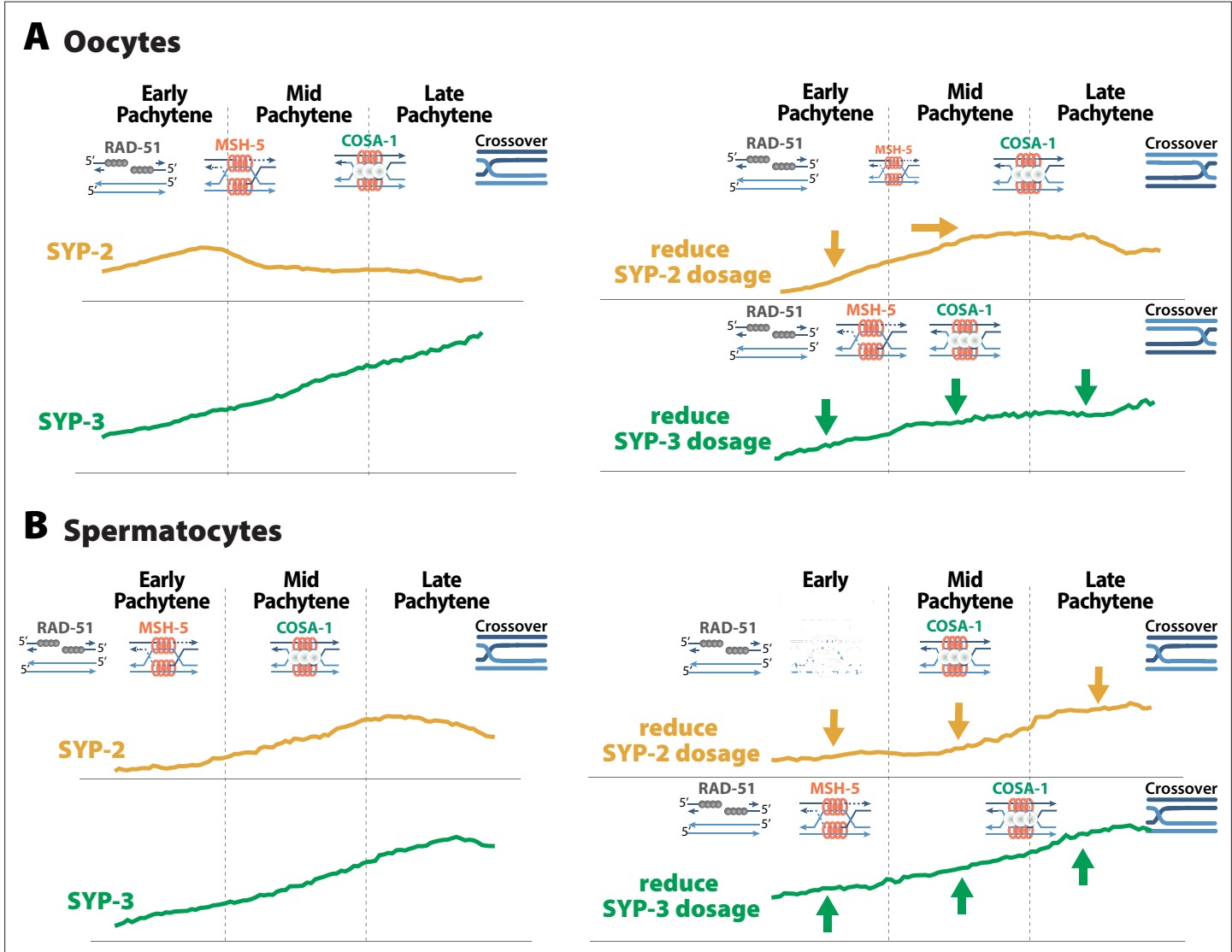

**Figure 9.** SYP dosage influences the sexually dimorphic regulation of recombination. (**A**) In oocytes, SYP-2 amounts with the synaptonemal complex (SC) are critical for the proper formation and/or maintenance of joint molecules stabilized by MSH-5. Reducing the amount of SYP-2 dosage causes decreases in the amount of SYP-2 in early pachytene and shifts the peak amounts of SYP-2 toward mid/late pachytene. This alteration to SYP-2 composition within the SC causes a severe reduction in MSH-5. SYP-3 amounts in the SC are important for the proper timing of recombination. When the dosage of SYP-3 is reduced, SYP-3 composition within the SC is reduced throughout pachytene causing faster resolution of jointed molecules and faster designation of crossovers during pachytene. This ultimately causes changes in the recombination landscape where crossovers are more often positioned near the pairing center. (**B**) In spermatocytes, SYP-2 amounts are important for the maintenance of MSH-5 stabilized joint molecules. When SYP-2 dosage is reduced, the amount of SYP-2 in the SC is reduced and MSH-5 foci are rapidly lost either because they are resolved quickly or the stability of the joint molecules is compromised. SYP-3 amounts in spermatocytes also influence the timing of recombination, but when SYP-3 dosage is reduced SYP-3 amounts are increased rather than decreased. Thus, elevated SYP-3 levels in spermatocytes cause a delay in crossover designation in spermatocytes.

levels in the SC are dependent on recombination such that SYP protein dosage impacts the regulation of recombination. Our data support a model where SYP-2 and SYP-3 levels within the SC regulate the proper timing and execution of specific steps of recombination (*Figure 9*).

Our data suggest that the amount of SYP-2 within the SC of oocytes promotes the formation and/or maintenance of joint molecules (*Figure 9A*). *syp-2/+* oocytes incorporate less SYP-2 in the SC during meiotic stages when MSH-5-stabilized joint molecules are formed (*Figures 5 and 7*) and exhibit a severe reduction in MSH-5 foci (*Figure 5*). This relationship between the SC and joint molecules is SYP-2 specific, as a reduction in SYP-3 levels within an assembled SC did not result in reduced MSH-5 foci (*Figures 5 and 7*). We therefore suggest that a specific threshold of SYP-2 in the SC ensures the formation and/or stabilization of joint molecules.

Similar to oocytes, spermatocyte SYP-2 dosage is required for the maintenance of joint molecules marked by MSH-5 (*Figure 9B*). Unlike oocytes, reducing SYP-2 levels in spermatocytes did not alter the assembly of the SC (*Figure 4*) and is likely why MSH-5 foci are able to initially form near wild-type levels in early pachytene (*Figure 5*). However, the rapid loss of MSH-5 in spermatocytes during mid pachytene suggests that these joint molecules are either being rapidly resolved and designated for crossover recombination or destabilized, resulting in MSH-5 removal. We favor a model where the rapid loss of MSH-5 foci is underpinned by destabilization of joint molecules as the crossover designation marker COSA-1 is not prematurely loaded when SYP-2 dosage is reduced. This function of SYP-2 in the maintenance of joint molecules appears conserved between the sexes as *syp-2/+* spermatocytes initially form wild-type levels of MSH-5 foci that are rapidly lost (*Figure 5*). However, the specific threshold of SYP-2 required in spermatocytes to establish and/or stabilize joint molecules is significantly reduced compared to oocytes (*Figures 2, 5, and 7*). This discrepancy between the sexes suggests that there are mechanistic differences in how joint molecules are regulated between spermatocytes and oocytes. Thus, our work here adds to a growing body of work illustrating that spermatocytes process DSBs into crossovers differently than oocytes (*Brick et al., 2018*; *Checchi et al., 2014*; *Durand et al., 2022*; *Jaramillo-Lambert et al., 2007*; *Jaramillo-Lambert and Engebrecht, 2010*; *Li et al., 2020*).

We further propose that SYP-3 dosage in oocytes regulates the timing of recombination steps (*Figure 9A*). SYP-3 accumulates within the SC throughout pachytene as recombination intermediates are successively processed (*Figure 2*). Notably, MSH-5-marked joint molecules and COSA-1-marked crossover designation appear prematurely when the levels of SYP-3 in the SC are reduced (*Figures 5 and 7*). This acceleration in crossover designation coincide with a disproportionate formation of crossovers near the pairing center side of chromosome *II* (*Figure 6*). This change in crossover designation and positioning was also present in *syp-2/+*. However, we assert that this phenotype is likely underpinned by SYP-3 levels as SYP-3 incorporation is also reduced in *syp-2/+* (*Figure 7*). Thus, the timing of recombination events in the germline appears to be sensitive to the amounts of SYP-3. We suggest that SYP-3 incorporation is dynamically regulated in response to meiotic stresses to ensure that recombination is completed by the end of pachytene, which is spatially and temporally limited by the length of the gonad.

In spermatocytes, SYP-3 also regulates the timing of recombination events during pachytene. However, spermatocytes' SYP-3 incorporation is minimally impacted by *syp-3/+* heterozygosity (*Figure 9B*). We suggest that the relatively normal levels of SYP-3 in the *syp-3/+* spermatocyte SC explain the absence of crossover positioning defects (*Figure 7*). In fact, SYP-3 levels are subtly elevated in early pachytene in *syp-3/+* spermatocytes and coincide with a delay in crossover designation (*Figures 5 and 7*). As spermatogenesis and oogenesis operate on very different timescales (*Jaramillo-Lambert et al., 2007*) and the amount of SYP-3 in the SC of spermatocytes and oocytes differs, we raise the possibility that titrating the level of SYP-3 incorporation may function to regulate the timing of recombination events between the sexes.

## Spermatocytes regulate the SC differently than oocytes

Sex-comparative studies are critical to understand the differences in egg and sperm development. Here we demonstrate that the SC in *C. elegans* is sexually dimorphic. Intriguingly, the sex-specific differences in SYP-2 and SYP-3 dynamics within the SC suggest a difference in protein regulation between the sexes (*Figure 1*). The progressive stabilization of SC during pachytene in oocytes has been linked with phosphorylation of specific SYP proteins (*Nadarajan et al., 2017*; *Pattabiraman*

*et al., 2017*). Additionally, during oogenesis, many SYP proteins are known to be post-translationally modified, including SYP-2, in response to recombination and many of these modifications are critical for DSB formation and repair as well as the timing of SC assembly and disassembly (*Garcia-Muse et al., 2019*; *Láscarez-Lagunas et al., 2022*; *Nadarajan et al., 2017*; *Nadarajan et al., 2016*; *Sato-Carlton et al., 2018*). Since spermatocytes also displayed the same progressive stabilization of the SYP-2 and SYP-3 during pachytene as oocytes, it is possible that the same post-translational modifications of SYP proteins may also have similar functions in spermatocytes.

We found that SYP-2 was more dynamic in spermatocytes even at the 'stabilized' state in late pachytene (*Figure 1*). Thus, spermatocytes may not stabilize the SC to the same degree as oocytes. One reason for this could be to allow DSBs in late pachytene to be repaired with the homolog. In oocytes, the stabilization of the SC in late pachytene is thought to prevent the formation of more crossovers with the homolog. Spermatocytes have been shown to have differences in DNA repair, the number of crossovers, and the checkpoints that monitor repair events, and these differences may require a more dynamic SC (*Checchi et al., 2014*; *Gabdank and Fire, 2014*; *Jaramillo-Lambert et al., 2007*; *Jaramillo-Lambert and Engebrecht, 2010*; *Jaramillo-Lambert et al., 2010*; *Li et al., 2020*). Future studies are needed in spermatocytes to examine these relationships between SC dynamics, post-translational modification of the SC, and DNA repair outcomes.

The amount of SYP-2, SYP-3, and SYP-5 required to properly execute the same steps of recombination is significantly different between oocytes and spermatocytes (*Figures 8 and 9*). Oocytes require more of both SYP-2, SYP-3, and SYP-5 during early and mid pachytene than spermatocytes (*Figures 2 and 8*). One possibility for this difference is due to male worms having a hemizygous X chromosome, which does not normally assemble the SC (*Jaramillo-Lambert and Engebrecht, 2010*). This lack of SC on one chromosome should potentially reduce the amount of SC proteins we get from our analysis. If this were the case, then we would expect the SYP proteins to be reduced in spermatocytes along the entire length of pachytene. Instead, we observe SYP-2 levels in spermatocytes reach those of oocytes during late pachytene and SYP-6 levels are nearly identical between the sexes in early pachytene regions (*Figures 2 and 9*). Further, spermatocytes have increased SYP-2 turnover dynamics in late pachytene compared to oocyte SYP-2 (*Figure 1*), which may help facilitate the repair of DSBs in late pachytene since spermatocytes do not undergo checkpoint-mediated apoptosis (*Jaramillo-Lambert et al., 2010*). Thus, having a more dynamic SYP-2 may aid DSB repair when errors occur late in pachytene.

Our observation that reduced SYP-3 dosage triggers an increase in SYP-3 composition within the SC suggests that spermatocytes may be able to compensate for the loss of a functional copy of *syp-3*. Given that spermatocytes differ in the dynamics of protein turnover within the assembled SC, one way spermatocytes could alter SYP-3 levels is to alter the SYP-3 turnover rates within the SC. We demonstrate that spermatocytes do indeed have altered protein dynamics in the SC compared to oocytes (*Figure 1*). Therefore, sex-specific differences in SYP-2 and SYP-3 dynamics may regulate the composition of the SC during pachytene.

## Independent regulation of the SYPs

One common feature amongst SC proteins in nearly all SC-containing organisms is that most, if not all, of the central region proteins are dependent upon each other for assembly of the SC (reviewed in *Cahoon and Hawley, 2016*). In *C. elegans*, the SYP proteins are also dependent on each other for protein stability. Specifically, the result where depletion of one SYP leads to the degradation of the other SYPs has led to the assumption that the SYPs are all completely interdependent for assembly, stability, and function (*Colaiácovo et al., 2003*). Here we show that fluctuations in both SYP-2 and SYP-3 levels can not only differentially influence the amount of each other within the SC and other SYPs (*Figures 7 and 8*), but also each SYP protein can be regulated independently (*Figures 2, 3, 7, and 8*). Notably, we show that these differences in the proportion of SYP-2 and SYP-3 within the SC are directly involved in regulating specific steps of recombination. Additionally, some of the SYPs display similar accumulation patterns and respond similarly to changes in SYP protein levels during pachytene, such as SYP-5 and SYP-3 in oocytes (*Figures 7 and 8*). Thus, we hypothesize that each SYP in the SC maintains both SYP-dependent functions where they function together as a group (e.g., assembling the SC) and SYP-independent functions where they can individually or in smaller groups to influence other aspects of meiosis (e.g., regulating specific steps of recombination).

In worms and yeast, the SC appears to grow in width throughout pachytene, suggesting that the composition of proteins within the complex is highly dynamic (*Pattabiraman et al., 2017*; *Voelkel-Meiman et al., 2012*). Here we found that the incorporation of SYP-2 and SYP-3 is not proportional during pachytene (*Figure 1*). Further, our results suggest that recombination influences the pattern of SYP accumulation within the SC and that each of the SYPs is independently regulated within the SC (*Figure 3*). The ability to adjust the amount of SYP-2, SYP-3, SYP-5, and SYP-6 within the SC demonstrates flexibility in the requirements of SYPs to assemble the complex. SYP-2 is positioned in the very center of the SC and may be slightly more external on the complex to SYP-3, SYP-5, and SYP-6 (*Köhler et al., 2020*), which would facilitate regulation of SYP-2 levels without compromising the whole complex. Also, the proteins within this very central region of the SC play a critical role in recombination (*Gordon et al., 2021*; *Voelkel-Meiman et al., 2019*; *Voelkel-Meiman et al., 2015*; *Voelkel-Meiman et al., 2022*). Thus, these proteins in the center of the SC, such as SYP-2, may not have as strong of structural roles in the complex as those positioned broadly in the central region, such as SYP-3.

The sex-specific differences with SYP-6 localization and the sexually dimorphic response of SYP-5 and SYP-6 to SYP-2 and SYP-3 levels suggest that SYP-5 and SYP-6 have some individual sex-specific roles during meiosis. Null mutants of *syp-5* and *syp-6* in oocytes also indicate that there are shared and distinct functions between these proteins (*Hurlock et al., 2020*; *Zhang et al., 2020*). *syp-5* mutants have stronger defects in fertility and crossing over than *syp-6* mutants, thereby suggesting that SYP-5 may play a more significant role in oocyte recombination than that of SYP-6 (*Hurlock et al., 2020*; *Zhang et al., 2020*). However, the sex-specific retention of SYP-6 localization into late pachytene of spermatocytes suggests that SYP-6 has a spermatocyte-specific role in late pachytene (*Figure 8*). Since spermatocytes do not undergo checkpoint-mediated apoptosis (*Jaramillo-Lambert et al., 2010*), retaining SYP-6 late in pachytene may facilitate repair of any DSBs still present into late pachytene. Future studies examining the role of SYP-6 in spermatocyte recombination may reveal that the fertility defects in *syp-6* mutants are caused by defective sperm rather than oocytes.

Not only are the patterns of SYP-2 and SYP-3 accumulation in the SC different, but the dosage of each SYP protein can influence the amount of the other SYP proteins. Regardless of which SYP dosage was altered, the same change in SYP-2 and SYP-3 accumulation occurred individually for each sex. In oocytes, both SYP-2, SYP-3, and SYP-5 amounts decreased in response to reduced SYP-2 or SYP-3 (*Figure 7*). Whereas in spermatocytes reduced SYP-2 or SYP-3 decreased SYP-2 amounts, increased SYP-3 amounts, and did not change SYP-5 amounts (*Figures 7 and 8*). The exception to this was SYP-6, which increased upon reduced SYP-2 dosage and did not change significantly in response to altered SYP-3 dosage (*Figure 8*). One explanation for these sex-specific differences is that spermatocytes require a different stoichiometry of proteins in the SC compared to oocytes. In mice, the SC in oocytes is narrower than the SC between spermatocytes due to structural differences in the organization of proteins within the central element and the chromosome axis (*Agostinho et al., 2018*). While *C. elegans* does not have a defined central element based on electron micrographs of the SC, SYP-2 is located in the very center of the SC where the central element proteins are located in other organisms (*Cahoon and Hawley, 2016*). Future studies to determine the stoichiometric ratios of chromosome axis proteins may reveal that spermatocytes assemble an SC that is structurally different from oocytes.

## Methods

**Key resources table**

| Reagent type (species) or resource | Designation | Source or reference | Identifiers | Additional information |
|---|---|---|---|---|
| Gene (*Caenorhabditis elegans*) | *syp-2* | https://wormbase.org/species/c_elegans/gene/WBGene00006376#0-9f-10 | WormBase ID: WBGene00006376 | |
| Gene (*C. elegans*) | *syp-3* | https://wormbase.org/species/c_elegans/gene/WBGene00006377#0-9f-10 | WormBase ID: WBGene00006377 | |

*Continued on next page*

*Continued*

| Reagent type (species) or resource | Designation | Source or reference | Identifiers | Additional information |
|---|---|---|---|---|
| Gene (*C. elegans*) | *syp-5* | https://wormbase.org/species/c_elegans/gene/WBGene00021832#0-9f-10 | WormBase ID: WBGene00021832 | |
| Gene (*C. elegans*) | *syp-6* | https://wormbase.org/species/c_elegans/gene/WBGene00019002#0-9f-10 | WormBase ID: WBGene00019002 | |
| Gene (*C. elegans*) | *syp-1* | https://wormbase.org/species/c_elegans/gene/WBGene00006375#0-9f-10 | WormBase ID: WBGene00006375 | |
| Gene (*C. elegans*) | *dsb-2* | https://wormbase.org/species/c_elegans/gene/WBGene00194892#0-9f-10 | WormBase ID: WBGene00194892 | |
| Gene (*C. elegans*) | *rad-51* | https://wormbase.org/species/c_elegans/gene/WBGene00004297#0-9f-10 | WormBase ID: WBGene00004297 | |
| Gene (*C. elegans*) | *msh-5* | https://wormbase.org/species/c_elegans/gene/WBGene00003421#0-9f-10 | WormBase ID: WBGene00003421 | |
| Gene (*C. elegans*) | *cosa-1* | https://wormbase.org/species/c_elegans/gene/WBGene00022172#0-9f-10 | WormBase ID: WBGene00022172 | |
| Strain, strain background (*C. elegans*) | For *C. elegans* alleles and strain information, see strain table below ('*C. elegans* strains, genetics, CRISPR, and culture conditions') | This paper | | See strain table below in '*C. elegans* strains, genetics, CRISPR, and culture conditions' |
| Genetic reagent (*C. elegans*) | For details on CRISPR/Cas9, see '*C. elegans* strains, genetics, CRISPR, and culture conditions' | This paper | | CRISPR/Cas9 transgenics performed by InVivo Biosystems |
| Antibody | Anti-RAD-51 (chicken polyclonal) | *Kurhanewicz et al., 2020*; *Toraason et al., 2021* | | IF (1:1500) |
| Antibody | Anti-SYP-1 (rabbit polyclonal) | Gift from Nicola Silva lab | | IF (1:1000) |
| Antibody | Anti-DSB-2 (rabbit polyclonal) | *Rosu et al., 2013* | | IF (1:5000) |
| Antibody | Anti-OLLAS (rabbit polyclonal) | GenScript | Cat# A01658 | IF (1:1000) |
| Antibody | Anti-SUN-1 S8P (guinea pig polyclonal) | *Woglar et al., 2013* | | IF (1:700) |
| Antibody | Anti-Tubulin (mouse monoclonal) | Abcam | Cat# ab7291 | WB (1:1000) |
| Antibody | Alexa Fluor 488 anti-rabbit (goat polyclonal) | Thermo Fisher | Cat# A11034 | IF (1:200) |
| Antibody | Alexa Fluor 488 anti-chicken (goat polyclonal) | Thermo Fisher | Cat# A11039 | IF (1:200) |
| Antibody | Alexa Fluor 555 anti-rabbit (goat polyclonal) | Thermo Fisher | Cat# A21428 | IF (1:200) |
| Antibody | Anti-GFP booster-488 (nanobody) | Chromotek | Cat# gb2AF488-50 | IF (1:200) |
| Antibody | Alexa Fluor 488 anti-guinea pig (goat polyclonal) | Thermo Fisher | Cat# A11073 | IF (1:200) |
| Antibody | IRDye 680 anti-mouse (donkey polyclonal) | LI-COR | Cat# 926-68072 | WB (1:1000) |
| Antibody | IRDye 800CW anti-rabbit (donkey polyclonal) | LI-COR | Cat# 926-32213 | WB (1:1000) |

*Continued on next page*

*Continued*

| Reagent type (species) or resource | Designation | Source or reference | Identifiers | Additional information |
|---|---|---|---|---|
| Sequence-based reagent | CRISPR primers are in **Supplementary file 1** | This paper | PCR primers | **Supplementary file 1** |
| Sequence-based reagent | For details on SNP recombination mapping primers, see 'SNP recombination mapping' and **Supplementary file 2** | This paper | PCR primers | **Supplementary file 2** |
| Chemical compound, drug | Serotonin | Sigma-Aldrich | Cat# H7752 | (25 mM) |
| Chemical compound, drug | Tricaine (ethyl 3-aminobenzoate methanesulfonate) | Sigma-Aldrich | Cat# E10521-50G | (0.08% w/v) |
| Chemical compound, drug | Tetramisole hydrochloride | Sigma-Aldrich | Cat# T1512-10G | (0.008% w/v) |
| Chemical compound, drug | Agarose | Invitrogen | Cat# 16500500 | (7–9% w/v) |
| Chemical compound, drug | Naphthaleneacetic acid (K-NAA, auxin) | PhytoTechnology Laboratories | Cat# N610 | (1 mM and 10 mM) |
| Software, algorithm | Whole Gonad Analysis (R script) | **Toraason et al., 2021** | https://github.com/libudalab/Gonad-Analysis-Pipeline | |
| Software, algorithm | Prism 10 | GraphPad | https://www.graphpad.com/features | |
| Software, algorithm | Imaris 9 | Oxford Instruments | https://imaris.oxinst.com/products | |
| Software, algorithm | FIJI plug in – 'stackRegJ' | https://research.stowers.org/imagejplugins/ | | |
| Software, algorithm | FIJI plug in – Stitcher | **Preibisch et al., 2009** | https://imagej.net/plugins/image-stitching | |
| Software, algorithm | FIJI | **Schindelin et al., 2012** | https://imagej.net/software/fiji/ | |
| Other | Low fluorescence PVDF membranes | Thermo Fisher | Cat# 22860 | |
| Other | Vectashield | VWR | Cat# 101098-042 | |
| Other | DAPI stain | Invitrogen | Cat# D1306 | (2 µg/mL) |

## *C. elegans* strains, genetics, CRISPR, and culture conditions

All strains were generated from the N2 background and were maintained and crossed at 20°C under standard conditions on nematode growth media (NGM) with lawns of *Escherichia coli*. InVivo Biosystems tagged the C-terminus of SYP-3 with a piRNA-optimized mCherry using CRISPR/Cas9. The CRISPR homology-directed repair template was constructed containing at least 500 base pairs of homology on either side of the insertion site at the SYP-3 locus. A small region of DNA was recoded section at the sgRNA site to avoid Cas9 cutting the template and mCherry was attached to SYP-3 with a glycine serine linker (GGSGGGGS). These repair constructs were synthesized into plasmids and injected into *unc-119(ed3)* mutant worms with two sgRNAs. Successful CRISPR/Cas9 integrations were screened using a *loxP* flanked *unc-119* rescue transgene, which was inserted into an intron of *syp-3* and removed following successful PCR confirmation of the integration by injecting Cre recombinase (**Dickinson et al., 2013**). All sequences and screening primers for the CRISPR/Cas9 tagging of SYP-3 are in **Supplementary file 1**. CRISPR/Cas9 worm lines were backcrossed to N2 worms three times, and loss of *unc-119(ed3)* mutation was confirmed by PCR before processing with any strain construction.

The following strains were used in this study:

N2: Bristol wild-type strain.

CB4856: Hawaiian wild-type strain.

DLW114: *unc-18(knu969[unc-18::AID*]) X. reSi7 [rgef-1p::TIR1::F2A::mTagBFP2::AID*::NLS::tbb-2 3'UTR] I.*

DLW118: *unc-18(knu969[unc-18::AID*]) X. reSi7 [rgef-1p::TIR1::F2A::mTagBFP2::NLS::AID*::tbb-2 3'UTR] I. GFP::syp-2 V.*

DLW119: *syp-3(knu999[mCherry::syp-3]) II.*

DLW128: *unc-18(knu969[unc-18::AID*]) X. reSi7 [rgef-1p::TIR1::F2A::mTagBFP2::NLS::AID*::tbb-2 3'UTR] syp-3(knu999[mCherry::syp-3]) I.*

DLW160: *unc-18(knu969[unc-18::AID*]) X. reSi7 [rgef-1p::TIR1::F2A::mTagBFP2::AID*::NLS::tbb-2 3'UTR] syp-3(knu999[mCherry::syp-3]) I. cosa-1(tm3298)/sC1(s2023) [dpy-1(s2170) umnIs41] III. GFP::syp-2 V.*

DLW163: *unc-18(knu969[unc-18::AID*]) X. reSi7 [rgef-1p::TIR1::F2A::mTagBFP2::AID*::NLS::tbb-2 3'UTR]. spo-11(me44)/nT1 [qIs51] IV. GFP::syp-2/nT1 [qIs51] V.*

DLW188: *syp-2(ok307)/tmC16 [unc-60(tmIs1210)] V.*

DLW190: *syp-3(ok785)/ tmC18 [dpy-5(tmIs1200)] I.*

DLW192: *unc-18(knu969[unc-18::AID*]) X. reSi7 [rgef-1p::TIR1::F2A::mTagBFP2::NLS::AID*::tbb-2 3'UTR] syp-3(knu999[mCherry::syp-3]) I. spo-11(me44)/nT1 [qIs51] IV. GFP::syp-2/nT1 [qIs51] V.*

DLW193: *unc-18(knu969[unc-18::AID*]) X. reSi7 [rgef-1p::TIR1::F2A::mTagBFP2::NLS::AID*::tbb-2 3'UTR] syp-3(knu999[mCherry::syp-3]) I. GFP::syp-2 V.*

DLW195: *meIs8[unc-119(+) pie-1promoter::gfp::cosa-1] II. syp-2(ok307)/tmC16 [unc-60(tmIs1210)] V.*

DLW196: *meIs8[unc-119(+) pie-1promoter::gfp::cosa-1] II. syp-3(ok785)/ tmC18 [dpy-5(tmIs1200)] I.*

DLW197: *unc-18(knu969[unc-18::AID*]) X. reSi7 [rgef-1p::TIR1::F2A::mTagBFP2::NLS::AID*::tbb-2 3'UTR] syp-3(knu999[mCherry::syp-3]) I. syp-2(ok307)/tmC16 [unc-60(tmIs1210)] V.*

DLW198: *unc-18(knu969[unc-18::AID*]) X. reSi7 [rgef-1p::TIR1::F2A::mTagBFP2::NLS::AID*::tbb-2 3'UTR] syp-3(ok785)/hT2 [bli-4(e937) let-?(q782) qIs48] (I;III). GFP::syp-2 V.*

DLW208: *msh-5[ddr22(GFP::msh-5)] IV; syp-2(ok307)/tmC16 [unc-60(tmIs1210)] V.*

DLW209: *syp-3(ok785)/ tmC18 [dpy-5(tmIs1200)] I; msh-5[ddr22(GFP::msh-5)] IV.*

DLW211: *cosa-1[ddr12(OLLAS::cosa-1)] III; syp-2(ok307)/tmC16 [unc-60(tmIs1210)] V.*

DLW212: *syp-3(ok785)/ tmC18 [dpy-5(tmIs1200)] I; cosa-1[ddr12(OLLAS::cosa-1)] III.*

AV630: *meIs8[unc-119(+) pie-1promoter::gfp::cosa-1] II.*

NSV97: *cosa-1[ddr12(OLLAS::cosa-1)] III.*

NSV129: *msh-5[ddr22(GFP::msh-5)] IV.*

DLW241: *unc-18(knu969[unc-18::AID*], knu1118 [rgef-1p::TIR1::F2A::mTagBFP2::NLS::AID::tbb-2 3'UTR downstream of unc-18::AID]) X. syp-5::gfp(cac4) I.*

DLW242: *unc-18(knu969[unc-18::AID*], knu1118 [rgef-1p::TIR1::F2A::mTagBFP2::NLS::AID::tbb-2 3'UTR downstream of unc-18::AID]) X. syp-6::gfp(cac5) I.*

DLW247: *unc-18(knu969[unc-18::AID*], knu1118 [rgef-1p::TIR1::F2A::mTagBFP2::NLS::AID::tbb-2 3'UTR downstream of unc-18::AID]) X. syp-5::gfp(cac4) I. syp-2(ok307)/tmC16 [unc-60(tmIs1210)] V.*

DLW248: *unc-18(knu969[unc-18::AID*], knu1118 [rgef-1p::TIR1::F2A::mTagBFP2::NLS::AID::tbb-2 3'UTR downstream of unc-18::AID]) X. syp-6::gfp(cac5) I. syp-2(ok307)/tmC16 [unc-60(tmIs1210)] V.*

DLW249: *unc-18(knu969[unc-18::AID*], knu1118 [rgef-1p::TIR1::F2A::mTagBFP2::NLS::AID::tbb-2 3'UTR downstream of unc-18::AID]) X. syp-5::gfp(cac4) syp-3(ok785)/ hT2 I.*

DLW250: *unc-18(knu969[unc-18::AID*], knu1118 [rgef-1p::TIR1::F2A::mTagBFP2::NLS::AID::tbb-2 3'UTR downstream of unc-18::AID]) X. syp-6::gfp(cac5) syp-3(ok785)/tmC18 [dpy-5(tmIs1200)] I.*

## Microscopy

Worms were mounted for all live imaging studies using our auxin-inducible conditional paralysis method, which is described in *Cahoon and Libuda, 2021*. Briefly, young adult worms (18–24 hr post L4) from a parental generation that was grown on NGM with either 1 mM auxin (for oocyte studies) or

10 mM auxin (for spermatocyte studies) were picked into 1 μL drop of live imaging media (M9 media with 25 mM serotonin [Sigma-Aldrich, Cat# H7752, *Rog and Dernburg, 2015*], 0.08% tricaine [ethyl 3-aminobenzoate methanesulfonate; Sigma-Aldrich, Cat# E10521-50G], 0.008% tetramisole hydrochloride [Sigma-Aldrich, Cat# T1512-10G]) and either 1 mM or 10 mM auxin (naphthaleneacetic acid [K-NAA], PhytoTechnology Laboratories, Cat# N610; *Martinez et al., 2020*) on a 22 × 40 mm (no. 1.5) coverslip. (Note: we found that poly-lysine treating the coverslips was not necessary for immobilization of the worms in most cases as long as the liquid under the agarose pad is minimal.) Also, 7–9% agarose pads (Invitrogen, Cat# 16500500) were gently placed over the top of the worms and excess liquid was wicked away using Whatman paper. A microscope slide was adhered to the agarose pad worm coverslip sandwich using a ring of Vaseline around the pad. Worms were then imaged using the setting described below. For both SC intensity and photobleaching studies, worms were imaged immediately following mounting and worms were only kept mounted for a max of 1 hr even though worms can survive being mounted for 2–3 hr (*Cahoon and Libuda, 2021*).

We did note that previous studies displayed higher fractions of SYP-3 recovery, which is likely caused by our studies not upshifting the worms to 25°C overnight (*Nadarajan et al., 2017*; *Pattabiraman et al., 2017*; *Rog et al., 2017*). Shifting the worms to 25°C is known to cause significant elevations in meiotic gene expression and is used to enhance expression of fluorescently tagged meiotic proteins (*Pattabiraman et al., 2017*; *Song et al., 2010*; *Yokoo et al., 2012*). Notably, we found that in recombination-deficient mutants, such as *cosa-1*, the elevated levels of GFP::SYP-2 protein in the SC with the elevation in protein expression that comes at 25°C caused large aggregates to form in mid/late pachytene that would persist into diakinesis (*Figure 3—figure supplement 3*). When the worms were only grown at 20°C without any 25°C upshift, these aggregates did not form. Thus, none of the worms for the studies in this paper were placed at 25°C overnight. Nevertheless, the trends appear the same for progressive stabilization with both GFP::SYP-3 and mCherry::SYP-3 (*Figure 1*, *Figure 1—figure supplement 2*), suggesting that mCherry and GFP fluorescent tags of SYP-3 do not appear to illicit differential dynamics (*Nadarajan et al., 2017*; *Pattabiraman et al., 2017*; *Rog et al., 2017*).

All live imaging studies of SYP-2::GFP, mCherry::SYP-3, SYP-5::GFP, and SYP-6::GFP were imaged on a Nikon CSU SoRa Spinning Disk Microscope with a ×60 water lens/N.A. 1.2 using a Z-step size of 0.3 μm. For SC intensity quantifications, the laser power and exposure times were kept consistent for all genotypes. All GFP::SYP-2, SYP-5::GFP, and SYP-6::GFP were imaged using the 488 laser at 16% power and 500 ms exposure time. All mCherry::SYP-3 imaging used the 561 laser at 25% power and 700 ms exposure time. Additionally, only the bottom half of the germline closest to the coverslip was imaged and germlines were not imaged if the position of the mounted worm caused the gut to cover germline or moved parts of the germline deeper into the worm.

The FRAP studies were performed as described in *Pattabiraman et al., 2017* with minor changes. Briefly, a Z-stack was taken prior to photobleaching to obtain a pre-bleach image. Then, a region of interest defined by the point tool in Elements was photobleached. A timelapse was started immediately post-photobleaching with images captured every 5 min for 35 min to monitor the fluorescence recovery. Pilot experiments showed that the 35 min timepoint displayed the highest recovery fluorescence observed before the signal plateaued. So, to minimize photobleaching and phototoxicity effects, we concluded the recovery timelapses at 35 min. For photobleaching small regions of GFP::SYP-2 or mCherry::SYP-3, a 405 laser was used with 1–5% laser power and 10–30 ms exposure depending on the germline location of the nucleus and the tagged protein with GFP::SYP-2 requiring less laser power to photobleach than mCherry::SYP-3. Previous studies showed that immobilized worms without serotonin have significantly diminished or absent chromosome motion, and the loss of this motion does not impair SC recovery dynamics (*Pattabiraman et al., 2017*; *Rog and Dernburg, 2015*; *Rog et al., 2017*). Since the addition of chromosome motion makes the FRAP recovery analysis very challenging, we did not include serotonin to minimize the motion of the chromosomes and to allow for better tracking of the photobleached SC region within each nucleus during the recovery timelapse.

Immunofluorescence slides of fixed gonad were imaged on a GE DeltaVision microscope with a ×63/N.A. 1.42 lens and 1.5× optivar at 1024 × 1024 pixel dimensions. Images were acquired using 0.2 μm Z-step size and deconvolved with softWoRx deconvolution software.

## Immunohistochemisty

Immunofluorescence was performed as described in *Cahoon and Libuda, 2021*; *Libuda et al., 2013*. Briefly, gonads were dissected in egg buffer with 0.1% Tween20 onto VWR Superfrost Plus slides from 18 to 24 hr post L4 worms. Dissected gonads were fixed in 5% paraformaldehyde for 5 min, flash frozen in liquid nitrogen, and fixed for 1 min in 100% methanol at −20°C. Slides were washed three times in PBS + 0.1% Tween20 (PBST) for 5 min each and incubated in block (0.7% bovine serum albumin in PBST) for 1 hr. Primary antibodies (chicken anti-RAD-51, 1:1500 [*Kurhanewicz et al., 2020*; *Toraason et al., 2021*]; rabbit anti-SYP-1, 1:1000 [gift from Silva Lab]; rabbit anti-DSB-2 [*Rosu et al., 2013*]; 1:5000; rabbit anti-OLLAS 1:1000 [GenScript, A01658]; guinea pig anti-SUN-1 S8P, 1:700 [*Woglar et al., 2013*]) were added and incubated overnight in a humid chamber with a parafilm cover. Slides were then washed three times in PBST for 10 min each and incubated with secondary antibodies (goat anti-rabbit Alexa Fluor 488, Thermo Fisher, Cat# A11034; goat anti-chicken Alexa Fluor488, Thermo Fisher, Cat# A11039; goat anti-rabbit Alexa Fluor 555, Thermo Fisher, Cat# A21428; GFP booster, Chromotek, gb2AF488-50; goat anti-guinea pig Alex Fluor 488, Thermo Fisher, Cat# A11073) at 1:200 dilution for 2 hr in a humid chamber with a parafilm cover. Slides were washed two times in PBST then incubated with 2 µg/mL DAPI for 15–20 min in a humid chamber. Prior to mounting, slides were washed once more in PBST for 10 min and mounted using Vectashield with a 22 × 22 mm coverslip (no. 1.5). Slides were sealed with nail polish and stored at 4°C until imaged. GFP::MSH-5 slides were imaged within 24–48 hr of mounting due to significant signal loss in the GFP::MSH-5 staining if the slide were stored longer.

## Image analysis and quantification

### FRAP quantification

The quantification of fluorescence recovery of GFP::SYP-2 and mCherry::SYP-3 was determined using FIJI. All photobleaching movies were first stabilized using the FIJI plugin 'StackRegJ' (https://research.stowers.org/imagejplugins/) to reduce the nuclear and worm motion in the germline. Photobleached nuclei were cropped to exclude as much extra z volume outside the size of the nucleus as possible. Then, these nuclei were sum intensity z-projected and the fluorescence intensity of the photobleached region was monitored using the rectangle tool. A small box was drawn on the segment of SC that will be photobleached, and through each frame of the timelapse the fluorescence intensity was recorded to obtain pre-bleach, bleach, and post-bleach fluorescence intensity values for a total of 35 min. Similar to *Pattabiraman et al., 2017*, we also excluded any nucleus that rotated or shifted in such a way that the photobleached SC segment could not be tracked between frames of the timelapse.

Nuclei in early pachytene were defined by being within the first 5–6 rows of pachytene, and nuclei in late pachytene were defined by being within the last 5–6 rows of pachytene. Mid pachytene nuclei were selected by being located within the middle region of pachytene. At each of these regions, the fluorescent intensity of three background regions of interest was determined per germline and averaged together to give the average background intensity. The average background intensity was subtracted from the fluorescence intensity of the photobleached SC segment. Additionally, the FRAP data from each SC segment was normalized such that the segment intensity pre-photobleach was 1 and the intensity immediately post-photobleached was 0. This allows us to determine the fraction of fluorescence intensity of each SYP protein that recovered following 35 min post-photobleaching. For oocytes, 3–5 germlines were used for GFP::SYP-2 and mCherry::SYP-3 analysis. For spermatocytes 3–6, germlines were used for GFP::SYP-2 and mCherry::SYP-3 analysis. For both sexes, 8–11 nuclei were analyzed in each region of pachytene and the specific n values in each region are reported in the figure legend. All images have been sum intensity projected and slightly adjusted for brightness and contrast. Additionally, any brightness and contrast adjustments made to oocyte images were also applied to spermatocyte images.

### SC intensity quantification

The quantification of GFP::SYP-2, mCherry::SYP-3, SYP-5::GFP, and SYP-6::GFP was performed using Imaris (Oxford Instruments) in combination with our whole gonad analysis, described in *Toraason et al., 2021*. The assembled, chromatin-associated SYP signal in each nucleus was surfaced in Imaris to obtain the sum intensity and volume of the assembled. The start of pachytene was defined by the first row that did not contain more the 1–2 nuclei of transition zone nuclei (nuclei with DNA in a

polarized or 'crescent'-shaped morphology) and full-length SC. We assessed nuclei shape using both the nuclear fluorescent haze produced by unassembled fluorescently tagged SYPs and the assembled chromatin-associated SC. The end of pachytene was defined by the last row that contained all pachytene nuclei with the occasional single diplotene nucleus. The pachytene region was then equally divided into three zones based on the length of this region within the germline to generate early pachytene, mid pachytene, and late pachytene. These criteria were used for establishing the early pachytene, mid pachytene, and late pachytene in both hermaphrodite and male germlines. Nuclei were excluded from the analysis if they were not in a single layer on the bottom half of the imaged germline rachis due to an intensity decrease caused by higher amounts of light scatter from being deeper in germline.

Our whole gonad analysis was used to align GFP::SYP-2, mCherry::SYP-3, SYP-5::GFP, and SYP-6::GFP surfaced nuclei along the germline length (*Toraason et al., 2021*). Each nucleus was then normalized by its volume to determine the normalized sum intensity of each nucleus during pachytene. The length of pachytene was also normalized per germline from 0 (early pachytene) to 1 (late pachytene). Since SYP-6 disassembles in oocytes prior to the end of pachytene, SYP-6::GFP was normalized to the SYP-6::GFP distance, which was measured from early pachytene (0) to the end of the SYP-6::GFP signal (1) for each germline. To calculate the average and standard deviation of the normalized SYP intensity of each nucleus during pachytene, we binned the data using a sliding window of 0.01. Also, 7–12 germlines were analyzed for all genotypes and both sexes. During the course of this study, we discovered that all images from May 2022 up to September 2022 needed to be corrected for a 15% drop in the power of the 561 nm laser. This correction was applied to the sum intensity of mCherry::SYP-3 for all genotypes imaged during this time period, which included *spo-11* oocytes and spermatocytes, *syp-2/+* oocytes and spermatocytes, *cosa-1* spermatocytes, *syp-3/+* oocytes and spermatocytes, and wild-type oocytes and spermatocytes. The number of nuclei analyzed within early, mid, and late pachytene in each genotype is reported in as source data for *Figures 2, 3, 7 and 8*. All images have been sum intensity projected and slightly adjusted for brightness and contrast with the same settings between mutants and sexes.

## SC length quantification

SC length measurements were determined on deconvolved DeltaVision images in Imaris using the filament tracer tool. Each chromosome within a nucleus was traced following the SYP-1 signal. If all six chromosomes could not be traced, then that nucleus was excluded from the analysis. Nuclei in early pachytene were defined by being within the first 5–6 rows of pachytene, and nuclei in late pachytene were defined by being within the last 5–6 rows of pachytene. Mid pachytene nuclei were selected by being located within the middle region of pachytene. For spermatocyte nuclei with six SC tracks, the smallest trace length was removed from the analysis because we inferred it to be the hemizygous *X* chromosome inappropriately assembled in the SC (*Jaramillo-Lambert and Engebrecht, 2010*). For oocytes, 10 nuclei in early pachytene were traced, 10 nuclei in mid pachytene were traced, and 12 nuclei in late pachytene were traced. For spermatocytes, 11 nuclei in early pachytene were traced, 12 nuclei in mid pachytene were traced, and 13 nuclei in late pachytene were traced.

## Germline measurement quantifications of transition zone, DSB-2 staining, SUN-1 S8P, and SYP-1 assembly

Germline measurement quantification was performed using Imaris in combination with our whole gonad analysis protocol, described in *Toraason et al., 2021*. Imaged gonads were stitched together using the FIJI (NIH) plugin Stitcher (*Preibisch et al., 2009*) and using the measurement tool in Imaris. The positions of points along the germline were recorded by marking specific regions indicating the start of the germline at the pre-meiotic tip, start of the transition zone, start of DSB-2 staining, start of SUN-1 S8P, start of SYP-1 assembly, end of transition zone, end of DSB-2 zone, end of SUN-1 S8P, end of SYP-1 assembly, end of SYP-1 zone, end of pachytene, last nuclei with SUN-1 S8P, and end of straggler DSB-2 nuclei. DAPI morphology was used to determine the start and end of the transition and pachytene. From these recorded point positions, we calculated the length of (1) the transition zone, (2) SYP-1 assembly zone, (3) full-length SYP-1 zone, (4) SUN-1 S8P zone, (5) last nucleus with SUN-1 S8P zone, (6) DSB-2 zone, (7) DSB-2 straggler nuclei zone, and (8) pachytene. The start of the transition zone and pachytene was defined by the first row that did not contain more than 1–2 nuclei of

either pre-meiotic nuclei (compact nuclei) or transition zone nuclei (nuclei with DNA in a polarized or 'crescent'-shaped morphology), respectively. The end of pachytene was defined by the last row that contained all pachytene nuclei with the occasional single diplotene nucleus. The start of DSB-2 and SUN-1 S8P was determined by the position where the staining of each antibody began in a majority of the nuclei within a row, and the end of DSB-2 and SUN-1 S8P staining was determined by the position where the staining of each antibody was largely absent from a majority of the nuclei. The end of the DSB-2 straggler nuclei and SUN-1 S8P last nucleus was defined by the last nucleus in the germline with bright DSB-2 staining or SUN-1 S8P staining, respectively. The start of the SYP-1 assembly zone was defined by the germline position where small linear fragments of SYP-1 were observed, and the end of the SYP-1 assembly zone was defined by the germline position where all the nuclei in a row had full-length SYP-1 with only 1–2 discontinuities. The end of the SYP-1 zone was determined by the region where SYP-1 began to disassemble at the end of pachytene. The germline length was normalized per germline where 0 was the start of the germline at the pre-meiotic tip and 1 was the end of late pachytene. The number of germlines analyzed in each experiment is reported in the figure legends. All images have been max intensity projected and slightly adjusted for brightness and contrast.

## RAD-51, MSH-5, and COSA-1 quantification

Imaged gonads were stitched together using the FIJI (NIH) plugin Stitcher (*Preibisch et al., 2009*) and analyzed in Imaris as described in *Toraason et al., 2021* with minor changes. Each gonad from the start of pachytene through the end of pachytene was analyzed for RAD-51, MSH-5, or COSA-1 foci per nucleus, which was determined by DAPI morphology. The start of pachytene was defined by the first row that did not contain more than 1–2 nuclei of transition zone nuclei (nuclei with DNA in a polarized or 'crescent'-shaped morphology). The end of pachytene was defined by the last row that contained all pachytene nuclei with the occasional single diplotene nucleus. The pachytene region was then equally divided into three zones based on the length of this region within the germline to generate early pachytene, mid pachytene, and late pachytene. The length of pachytene was also normalized per germline from 0 (early pachytene) to 1 (late pachytene). These criteria were used for establishing the early pachytene, mid pachytene, and late pachytene in both hermaphrodite and male germlines. For RAD-51, MSH-5, and COSA-1 foci per nucleus, sliding window averages and standard error of the mean (SEM) were calculated using a 0.01 bin size. Also, 7–12 germlines were analyzed for all genotypes and both sexes. The number of nuclei analyzed within early, mid, and late pachytene in each genotype is reported in the figure legends for each plot in *Figure 5*. All images have been max intensity projected and slightly adjusted for brightness and contrast.

## Western blot analysis

For both male and hermaphrodite, 100 adult worms were picked and washed once with M9 before being boiled for 5–10 min in sample buffer with occasional vortexing. For all western blots, samples were run on SDS-PAGE and wet transferred to low fluorescence PVDF membranes (Thermo Fisher Cat# 22860). Membranes were blocked in 5% milk in TBS + 0.1% Tween20 (TBST) for 1 hr at room temperature. Primary antibodies (anti-SYP-2 rabbit 1:1000 [gift from Yumi Kim], anti-Tubulin mouse 1:1000 [Abcam Cat# ab7291]) were incubated and agitated on an orbital shaker overnight at 4°C in 5% milk. Blots were washed three times for 10 min with TBST and LI-COR secondary antibodies (IRDye 680 donkey anti-mouse [LI-COR Cat# 926-68072], IRDye 800CW donkey anti-rabbit [LI-COR Cat# 926-32213]) were incubated at 1:1000 in TBST at room temperature for 1 hr. Blots were washed twice for 10 min each and imaged using LI-COR Odyssey Fc.

Since male worms only have one germline, all hermaphrodite samples were diluted to adjust for the two germlines present in hermaphrodite worms. To determine the correct dilution factor, we performed a dilution series of the hermaphrodite sample and measured the sum intensity of the SYP-2 bands in each dilution using FIJI (*Figure 3—figure supplement 2*). Then, we subtracted the background intensity from each SYP-2 measurement and normalized each dilution by the average intensity of SYP-2 from two undiluted male samples (*Figure 3—figure supplement 2*). From the normalized dilution curve, we determined that a 40% dilution of the hermaphrodite sample equated to similar amounts of SYP-2 in the undiluted male samples. This 40% dilution was applied to all hermaphrodite samples in *Figure 3—figure supplement 2*. To quantify the amount of SYP-2 in each sample, we used FIJI to measure the sum intensity of the SYP-2 and loading control alpha-tubulin bands that

were both subtracted from a background intensity. Then, the background-subtracted SYP-2 intensity was normalized by the background-subtracted loading control alpha-tubulin intensity to generate a normalized SYP-2 intensity (*Figure 3—figure supplement 2*).

## Fertility assay

To assess hermaphrodite fertility, L4 hermaphrodite worms were placed onto new NGM plates and were transferred every 24 hr for a total of 2 d. To assess male fertility, single L4 male worms were mated to single L4 *fog-2* obligate females and transferred every 24 hr for a total of 2 d before permanently removing the parental worms. After 3 d of removing the parental worms, each plate was scored for dead eggs. Then, the following day (4 d post removing parental worms) each plate was scored for living hermaphrodite and male progeny (male progeny was not scored in male fertility assays). Additionally, any progeny with mutant Unc or Dpy phenotypes was also scored for five worms in each genotype. Also, 7–10 worms were assayed for fertility for each genotype.

## SNP recombination mapping

SNP recombination mapping of chromosome *II* and *X* was performed as described in *Bazan and Hillers, 2011* with minor changes. *syp-2(ok307)* and *syp-3(ok785)* were generated in Bristol (N2) backgrounds, and we PCR confirmed that both mutant strains carried all Bristol SNPs for both chromosomes assayed prior to mapping recombination. To generate Bristol/Hawaiian hybrids for mapping recombination, we crossed Bristol, *syp-2(ok307)* (DLW188), and *syp-3(ok785)* (DLW190) hermaphrodites to Hawaiian (CB4856) males. Then, 8–10 Bristol/Hawaiian hybrid L4 hermaphrodites were picked off the cross plates for oocyte recombination mapping and 10–15 Bristol/Hawaiian hybrid L4 males were picked off the cross plates for spermatocyte recombination. For oocyte recombination mapping, hybrid L4 hermaphrodites of each genotype were crossed to Bristol males, and male progeny were picked into 96-well plates for lysis and recombination PCR screening. For spermatocyte recombination mapping, hybrid L4 males of each genotype were crossed to Bristol hermaphrodites and male progeny were picked into 96-well plates for lysis and recombination PCR screening.

We used previously designed PCR primers and restriction digests to map six Bristol and Hawaiian SNPs on both chromosomes *II* and *X* (*Bazan and Hillers, 2011*). However, we were unable to get the primers to work for SNP E on chromosome *II*, so we redesigned new primers for this SNP that worked with the existing restriction digest for SNP identification at this genomic location. All SNP positions, PCR primers, restriction digests, and band sizes of the products for the Bristol or Hawaiian SNPs can be found in *Supplementary file 2*. The recombination frequency or map length in centiMorgans (cM) was calculated by taking the total number of crossovers identified in each interval divided by the total chromosomes scored multiplied by 100.

## Statistics

Statistical analysis of the FRAP data, SC length, SNP recombination mapping, and fertility assays was done using Prism. Mann–Whitney *U* tests adjusted for multiple comparisons using the Bonferroni–Dunn method were performed on the FRAP data. Kruskal–Wallis tests were performed on the SC lengths with corrections for multiple comparisons. Chi-squared tests were performed on the entire SNP recombination mapping distribution. Pairwise comparisons between recombination intervals were performed using Fisher's exact test. Chi-squared and Fisher's exact tests were performed on the fertility assays. Mann–Whitney *U* tests adjusted for multiple comparisons using the Bonferroni–Dunn method were performed on the SC intensity, quantification of RAD-51, MSH-5, and COSA-1, transition zone length, pachytene length, end of SYP-1 assembly zone, SUN-1 S8P zone, end of DSB-2 zone, and end of DSB-2 straggler zone using R. Each test used is indicated in the 'Results' section next to the reported p value, and all n values are reported in the figure legends.

## Acknowledgements

We thank the CGC for strains, which is funded by the NIH Office of Research Infrastructure Programs (P40 OD010440). We thank the members of the Libuda Lab for discussion and comments on the manuscript, the Nicola Silva lab for the SYP-1 antibody, and the Yumi Kim lab for the SYP-2 antibody. This work was supported by the National Institutes of Health R35GM128890 to DEL and a Jane Coffin

Childs Postdoctoral Fellowship and National Institutes of Health 1K99HD109505-01 to CKC. DEL is also a Searle Scholar and recipient of a March of Dimes Basil O'Connor Starter Scholar award.

## Additional information

### Funding

| Funder | Grant reference number | Author |
|---|---|---|
| National Institute of General Medical Sciences | R35GM128890 | Diana E Libuda |
| Eunice Kennedy Shriver National Institute of Child Health and Human Development | 1K99HD109505 | Cori K Cahoon |
| Jane Coffin Childs Memorial Fund for Medical Research | | Cori K Cahoon |

The funders had no role in study design, data collection and interpretation, or the decision to submit the work for publication.

### Author contributions

Cori K Cahoon, Conceptualization, Formal analysis, Supervision, Funding acquisition, Validation, Investigation, Visualization, Methodology, Writing - original draft, Writing - review and editing; Colette M Richter, Amelia E Dayton, Investigation; Diana E Libuda, Conceptualization, Resources, Supervision, Funding acquisition, Project administration, Writing - review and editing

### Author ORCIDs

Cori K Cahoon ⓘ http://orcid.org/0000-0002-7888-2838
Diana E Libuda ⓘ http://orcid.org/0000-0002-4944-1814

### Decision letter and Author response

Decision letter https://doi.org/10.7554/eLife.84538.sa1
Author response https://doi.org/10.7554/eLife.84538.sa2

## Additional files

### Supplementary files
• Supplementary file 1. CRISPR/Cas9 primer sequences.
• Supplementary file 2. SNP recombination mapping primer sequences.
• MDAR checklist

### Data availability
All strains developed as part of this study will be available at the CGC or are available upon request. All data generated or analyzed during this study are included in the manuscript and supporting files. Source data have been provided for the numerical values plotted in the following figures: Figure 1, Figure 1—figure supplement 3, Figure 2, Figure 3, Figure 3—figure supplement 2, Figure 4, Figure 4—figure supplement 1, Figure 5, Figure 5—figure supplement 3, Figure 6, Figure 6—figure supplement 1, Figure 7, and Figure 8.

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
