## [Editor Report]

This important article describes sex- and recombination-dependent dynamics of proteins in a meiosis-specific chromosome structure, the synaptonemal complex. The authors provide solid evidence for their conclusion by cytological analysis with proper quantification. The study is of great interest to researchers in the field of meiosis and chromosomes.

---

## [Decision Letter]

**Decision letter after peer review:**

Thank you for submitting your article "Sexual dimorphic regulation of recombination by the synaptonemal complex" for consideration by *eLife*. Your article has been reviewed by 3 peer reviewers, including Akira Shinohara as the Reviewing Editor and Reviewer #1, and the evaluation has been overseen by Molly Przeworski as the Senior Editor.

Essential revisions:

Based on reviews and discussions with reviewers, we do appreciate that your cytological and genetic analyses showing that two components of the central region of the synaptonemal complex (SC), SYP-2 and SYP-3, show sexually dimorphic dynamics and that the gene dosage of the two genes affects dynamics of SYP-2 and SYP-3 as well as dynamics and numbers of the recombination proteins, MSH-5 and COSA-1 distinctly in oocytes and spermatocytes. However, we are very sorry that we are not positive about the publication of the current version of the paper. Your paper is very descriptive and does not provide molecular insights into how oocytes and spermatocytes control the two SC components, SYP-2 and SYP-3 differentially and also how the gene dosage of the two components affects the dynamics of themselves as well as the recombination proteins.

To strengthen the conclusion of the paper, we asked you to address the following major points as well as comments from each reviewer. If you do not agree with our comments, please feel free to rebut them.

1. In this paper, the dynamics of SYP-2 and SYP-3, have been described. A previous study by Dernburg and her colleagues showed that the central region of the SC has liquid crystalline properties (*eLife*, 2017) by analyzing SYP-2 dynamics. In this regard, the different behaviour of SYP-3 looks interesting given that the two components are in the same compartment within a liquid crystalline matrix. It would be very nice to show/confirm the other components of the SC central region such as SYP-1, SYP-4, and SYP-5/6. As an extension, it would be nice to examine the haplo-inefficiency of SYP-2 and/or SYP-3 on the dynamics of the aforementioned SC components. Do those other SC components also display haploinsufficiency? This could at least be addressed with basic assays.

2. To address the sexual dimorphic properties of SYP-2 and SYP-3, it is very important to check the amount of proteins in both oocytes and spermatocytes. And if possible, it would be great to address the difference in post-translational modifications between the two sexes.

3. The authors used different tags (GFP for SYP-2 and mCherry for SYP-3) in the FRAP experiments. In the strict sense, it would be better to use the same tag for the analysis of dynamics. At least the authors have to show the different tags do not affect the dynamics of the proteins in gonads by experiments (or citing the paper).

4. It is important to re-write the text in a more concise way including shortening the Introduction part. Rather than mentioning P-values in the text, it would be nice to put these values in a different (supplemental) table or put it in the legend.

---

## [Author Response]

Essential revisions:Based on reviews and discussions with reviewers, we do appreciate that your cytological and genetic analyses showing that two components of the central region of the synaptonemal complex (SC), SYP-2 and SYP-3, show sexually dimorphic dynamics and that the gene dosage of the two genes affects dynamics of SYP-2 and SYP-3 as well as dynamics and numbers of the recombination proteins, MSH-5 and COSA-1 distinctly in oocytes and spermatocytes. However, we are very sorry that we are not positive about the publication of the current version of the paper. Your paper is very descriptive and does not provide molecular insights into how oocytes and spermatocytes control the two SC components, SYP-2 and SYP-3 differentially and also how the gene dosage of the two components affects the dynamics of themselves as well as the recombination proteins.To strengthen the conclusion of the paper, we asked you to address the following major points as well as comments from each reviewer. If you do not agree with our comments, please feel free to rebut them.1. In this paper, the dynamics of SYP-2 and SYP-3, have been described. A previous study by Dernburg and her colleagues showed that the central region of the SC has liquid crystalline properties (eLife, 2017) by analyzing SYP-2 dynamics. In this regard, the different behaviour of SYP-3 looks interesting given that the two components are in the same compartment within a liquid crystalline matrix. It would be very nice to show/confirm the other components of the SC central region such as SYP-1, SYP-4, and SYP-5/6. As an extension, it would be nice to examine the haplo-inefficiency of SYP-2 and/or SYP-3 on the dynamics of the aforementioned SC components. Do those other SC components also display haploinsufficiency? This could at least be addressed with basic assays.

We agree with the reviewers that assessing the composition of all SYPs in the SC is a great experiment to complement the SYP-2 and SYP-3 studies. Unfortunately, we were unable to do live imaging experiments with SYP-1 as multiple groups have found it is rendered nonfunctional when tagged on the N- or C-terminus with a fluorescent tag (Gao et al., 2016; Köhler et al., 2022; Subramaniam et al., 2018). Although we generated a GFP::SYP-4 strain, we found that SYP-4 is not fully functional in this strain, therefore we were unable use it in this study.

For SYP-5 and SYP-6, we are very grateful to the reviewers for suggesting these experiments as we found that both of these proteins exhibited sexually dimorphic phenotypes and responses to *syp-2* and *syp-3* haploinsufficiency. Similar to the other SYPs, SYP-5 amounts are reduced in spermatocytes compared to oocytes. Further, oocyte SYP-5 accumulations in the SC and responses to *syp-2* and *syp-3* haploinsufficiency was very similar to SYP-3 suggesting these proteins may be similarly regulated. Notably, SYP-6 is the only SYP protein to display sexually dimorphic localization. Spermatocytes retain SYP-6 through late pachytene, whereas in oocytes SYP-6 is largely absent from late pachytene nuclei. Additionally, SYP-6 responds differently than SYP-5 to *syp-2* and *syp-3* haploinsufficiency in each sex, thereby suggesting that SYP-5 and SYP-6 may have some non-redundant roles. Together, these experiments enhanced our conclusions about the sexually dimorphic regulation of the SC and are now included as new figures (Figure 8, Figure 8 —figure supplement 1, Figure 8 —figure supplement 2) and a new Results section (see “SYP dosage differentially influences SYP-5 and SYP-6 composition within the SC”). Further, parts of the Discussion section have been expanded to include the conclusions from (and discussion of) this new data.

2. To address the sexual dimorphic properties of SYP-2 and SYP-3, it is very important to check the amount of proteins in both oocytes and spermatocytes. And if possible, it would be great to address the difference in post-translational modifications between the two sexes.

We agree with the reviewers that measuring total protein amounts of both SYP-2 and SYP-3 would complement our studies in the manuscript. To this end, we detected SYP-2 by western blot for wild type, *syp-2/+, syp-3/+,* and *spo-11* adult males (spermatogenesis) and adult hermaphrodites (oogenesis) (Figure 3 —figure supplement 2). These experiments are now included in Figure 3 —figure supplement 2 with the experimental method described in a new Methods section (see “Western Blot Analysis”). Unfortunately, we and others in the field have been unable to efficiently detect SYP-3 on western blot both using antibodies to the protein and fluorescent tag (our lab; Yumi Kim personal communication; Smolikov *et al.*, 2007), therefore we were unable to assess SYP-3 levels via western blot.

3. The authors used different tags (GFP for SYP-2 and mCherry for SYP-3) in the FRAP experiments. In the strict sense, it would be better to use the same tag for the analysis of dynamics. At least the authors have to show the different tags do not affect the dynamics of the proteins in gonads by experiments (or citing the paper).

We understand the reviewers concerns about using two different fluorescent tags for the FRAP comparisons and have included text in the manuscript to directly address this point. Specifically, our data with mCherry::SYP-3 is very similar to published data using GFP::SYP-3. Both sexes showed the same overall trend of progressive stabilization of SYP-2 and SYP-3 throughout pachytene matching previous observations with the transgene GFP::SYP-3 (Figure 1, Figure 1 —figure supplement 1, Figure 1 —figure supplement 2) (Nadarajan *et al.,* 2017; Pattabiraman *et al.*, 2017; Rog *et al.*, 2017). We did note that these previous studies displayed higher fractions of SYP-3 recovery, which is likely caused by our studies not upshifting the worms to 25ºC overnight (Nadarajan *et al.*, 2017; Pattabiraman *et al.*, 2017; Rog *et al.*, 2017). Shifting the worms to 25ºC is known to cause significant elevations in meiotic gene expression and is used to enhance expression of fluorescently tagged meiotic proteins (Pattabiraman *et al.*, 2017; Song *et al.*, 2010; Yokoo *et al.*, 2012). Nevertheless, the trends appear the same for progressive stabilization with both GFP::SYP-3 and mCherry::SYP-3 suggesting that mCherry and GFP fluorescent tags of SYP-3 do not appear to illicit differential dynamics (Nadarajan *et al.*, 2017; Pattabiraman *et al.*, 2017; Rog *et al.*, 2017). We have added this clarification into the Methods “Microscopy” section and reference it in the main text .

4. It is important to re-write the text in a more concise way including shortening the Introduction part. Rather than mentioning P-values in the text, it would be nice to put these values in a different (supplemental) table or put it in the legend.

We have consolidated the text of the manuscript and moved some of the p values, averages, etc. to the figure legends or additional supplemental tables where we felt it was appropriate. To be accessible to all readers, we did keep some of the p values in the text and in the figures/legends as some readers focus more on reading the figures/legends while others focus on reading the text over the legends.